Resource

# Chromosomal barcodes for simultaneous tracking of near-isogenic bacterial strains in plant microbiota

Jana Ordon [1,4,8], Julien Thouin [1,8], Ryohei Thomas Nakano[1,5], Ka-Wai Ma[1,6], Pengfan Zhang[1,7], Bruno Huettel [1], Ruben Garrido-Oter [1,2,3] & Paul Schulze-Lefert [1,2] ✉

DNA-amplicon-based microbiota profiling can estimate species diversity and abundance but cannot resolve genetic differences within individuals of the same species. Here we report the development of modular bacterial tags (MoBacTags) encoding DNA barcodes that enable tracking of near-isogenic bacterial commensals in an array of complex microbiome communities. Chromosomally integrated DNA barcodes are then co-amplified with endogenous marker genes of the community by integrating corresponding primer binding sites into the barcode. We use this approach to assess the contributions of individual bacterial genes to *Arabidopsis thaliana* root microbiota establishment with synthetic communities that include MoBacTag-labelled strains of *Pseudomonas capeferrum*. Results show reduced root colonization for certain mutant strains with defects in gluconic-acid-mediated host immunosuppression, which would not be detected with traditional amplicon sequencing. Our work illustrates how MoBacTags can be applied to assess scaling of individual bacterial genetic determinants in the plant microbiota.

Plants are inhabited by taxonomically structured multi-kingdom microbial communities referred to as plant microbiota. Members of the microbiota provide beneficial services to the plant host[1], including mobilization of nutrients[2,3], indirect pathogen protection[4–6] and abiotic stress tolerance[7,8]. Quantitative and cultivation-independent analysis of these microbial communities using marker genes, such as the *16S rRNA* gene of bacteria or *internal transcribed spacer* (*ITS*) regions of fungi, typically relies on the detection of natural nucleotide polymorphisms in hypervariable regions of the markers, thereby defining distinct microbial taxa. Computational analysis of marker-gene-based amplicon DNA sequencing data generated using PCR with marker-gene-specific primers has shifted from clustering similar reads into operational taxonomic units to error-correction approaches that account for individual amplicon sequence variants[9–11]. Despite an increased taxonomic resolution of bacterial communities through classification by amplicon sequence variants, profiling based on *16S rRNA* is still unable to capture the true within-phylotype genetic variation of microbial communities. Strains with identical marker gene sequences may comprise a bacterial population with other polymorphic loci associated with beneficial or detrimental plant traits, as shown for Rhizobiales and *Pseudomonas* lineages[12–14]. In such cases, only cumulative relative *16S rRNA* abundances originating from multiple strains represented

[1]Department of Plant–Microbe Interactions, Max Planck Institute for Plant Breeding Research, Cologne, Germany. [2]Cluster of Excellence on Plant Sciences (CEPLAS), Max Planck Institute for Plant Breeding Research, Cologne, Germany. [3]Earlham Institute, Norwich, UK. [4]Present address: Institute of Plant Molecular Biology, University of Zurich, Zurich, Switzerland. [5]Present address: Department of Biological Sciences, Faculty of Science, Hokkaido University, Sapporo, Japan. [6]Present address: Institute of Plant and Microbial Biology, Academia Sinica, Taipei, Taiwan. [7]Present address: Innovative Genomics Institute (IGI), University of California, Berkeley, CA, USA. [8]These authors contributed equally: Jana Ordon, Julien Thouin. ✉e-mail: schlef@mpipz.mpg.de

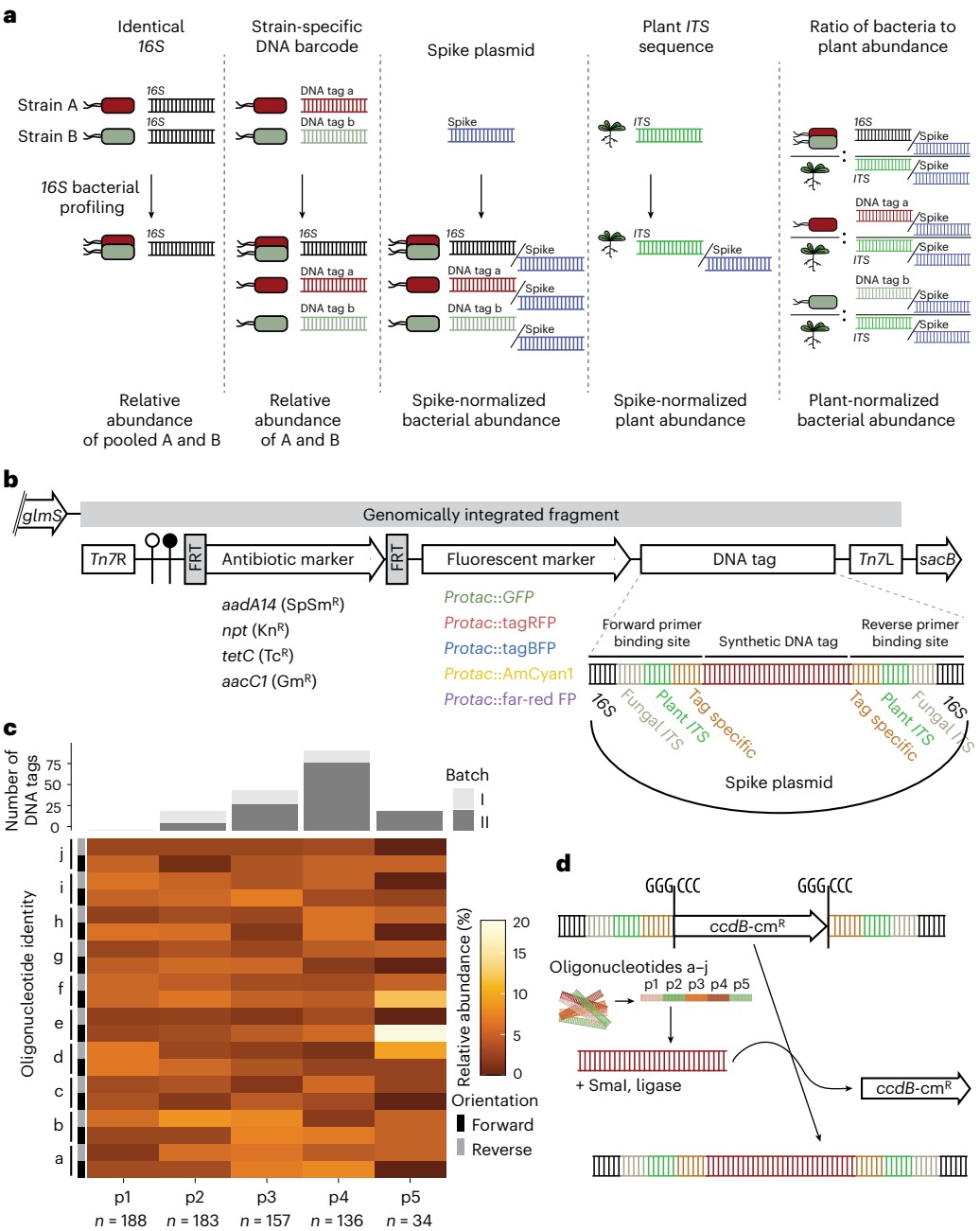

**Fig. 1 | Design and assembly strategy of MoBacTag vectors for simultaneous tracking of near-isogenic bacterial strains. a**, Principles of MoBacTag DNA barcodes for discriminating bacterial strains encoding identical *16S rRNA* sequences during amplicon sequencing for microbiota profiling and for quantifying microbial load using MoBacTag-based spike-in plasmids. **b**, Schematic representation of the MoBacTag. Each vector encodes 1 of the 20 possible combinations of the 4 antibiotics with the 5 fluorescent markers. The MoBacTag fragment (grey bar) is chromosomally integrated downstream of the *glucosamine-6-phosphate synthetase* (*glmS*) gene. The synthetic DNA tag is flanked by primer binding sites for amplification of bacterial *16S*, fungal *ITS*, plant *ITS* or barcode sequences. Plasmids encoding only the synthetic DNA tag were further used as spike plasmids. *Tn7*R and *Tn7*L, right and left *Tn7* borders, respectively; SpSm^R, spectinomycin and streptomycin resistance; Kn^R,

kanamycin resistance; Tc^R, tetracycline resistance; Gm^R, gentamycin resistance; Protac, synthetic bacterial hybrid promoter; *sacB*, levansucrase gene conferring sucrose sensitivity; tagBFP, blue fluorescent protein; AmCyan1, cyan fluorescent protein; far-red FP, far red fluorescent protein. The elements are not drawn to scale. **c**, Oligonucleotide composition of the synthetic DNA tags. The upper panel shows the number of oligonucleotides integrated into the DNA barcode tags. Grey colours represent two different assembly batches. The lower panel shows heat-map-visualized frequencies of individual oligonucleotides (a–j) in forward or reverse orientation at each position (p1, p2, p3, p4 and p5) of the array of ligated oligonucleotides within the DNA barcode tag. **d**, Integration strategy of preassembled DNA barcode tags into MoBacTag recipient vectors. Preassembled DNA barcode tags are inserted between primer binding sites in exchange for the dominant negative selection marker *ccdB*. cm^R, chloramphenicol resistance.

by a phylotype can be retrieved (Fig. 1a). This imposes limitations on functional microbiota studies, as the bacterial traits often vary in a strain-specific manner. Studies on genetic variation within a bacterial

phylotype are therefore typically limited to host mono-associations with cultured bacterial or fungal strains, and competition experiments with antibiotic markers[15], or depend on DNA sequencing of

strain-differentiating amplicons restricted to a particular taxon[16]. Furthermore, amplification of endogenous marker sequences does not allow differentiation between wild-type and mutant strains in a community context, limiting the application of microbial genetics in gnotobiotic systems.

Cellular barcodes can overcome resolution limitations by labelling strains or individual cells with unique DNA sequences, here referred to as DNA barcodes (Fig. 1a)[17]. DNA barcoding has been used to track cell lineages during experimental evolution[18], neuron dispersal[19], stem cell differentiation[20] or the development of drug resistance in cancer cells[21]. Furthermore, DNA barcodes chromosomally integrated at strain-specific neutral sites using, for example, homologous recombination or RNA-guided nucleases such as CRISPR–Cas9 are used to study host bottlenecks and subsequent tissue-specific population dynamics of bacterial pathogens in animal systems[22–24]. In plant–microbe interactions, DNA barcodes have been used mainly for screening bacterial mutant libraries based on a method called BarSeq[25–27] and to distinguish closely related *Pseudomonas* strains in the phyllosphere of *Arabidopsis thaliana*[28]. However, simultaneous profiling of taxonomically diverse microbial communities by amplicon sequencing of marker genes and of DNA barcodes of near-isogenic strains is not established in the field of vertebrate or plant microbiota studies.

Here we describe a modular bacterial tag (MoBacTag) tool to label a broad range of taxonomically distinct bacteria with a DNA barcode as well as a fluorescent tag. The DNA barcode allows discrimination of bacteria that cannot be distinguished when profiling microbial communities using conventional marker gene amplicon sequencing. As the DNA barcode is flanked by primer binding sites specific to *V5–V7 16S rRNA*, fungal *ITS*, plant *ITS* and barcodes, the abundances of barcoded bacteria can be determined simultaneously with abundances of unlabelled bacteria and fungi by amplicon sequencing of the corresponding marker genes. Furthermore, DNA-barcode-harbouring plasmids are used as spike DNA to estimate microbial load and calculate the ratio of plants to microbes. As a proof of principle, we use the MoBacTags to recapitulate the colonization defect of *Pseudomonas capeferrum* WCS358 *cyoB* and *pqqF* immunosuppressive mutants[29]. By simultaneously analysing DNA barcodes and *16S rRNA* sequences, we reveal an activity specific to the WCS358 *pqqF* mutant in community establishment, which is not *trans*-complemented by wild-type WCS358.

## Results

### Design of MoBacTag tools

MoBacTag tools have been designed for orientation-specific high-frequency insertion into bacterial chromosomes at the conserved *Tn7* attachment site[30] downstream of the *glmS* gene that is present in 98.6% (426 out of 432) of all bacterial draft genomes in the *Arabidopsis*-derived bacterial culture collection (*At*-R-SPHERE[31]; Extended Data Fig. 1)[32]. To show broad utility, we have labelled a total of 22 plant-derived bacterial strains from 9 different genera with MoBacTags by *Tn7*-mediated chromosomal integration (*Achromobacter*, Deinococcales, *Lysobacter*, *Neorhizobium*, *Pseudomonas*, *Pseudoxanthomonas*, *Rhizobium*, *Rhodanobacter*, *Xanthomonas*; Supplementary Table 1).

The chromosomally integrated fragment contains (1) the minimal *Tn7* elements (*Tn7*R, *Tn7*L)[33], (2) two terminator sequences, (3) an antibiotic marker flanked by yeast *Flp* recombinase recognition sites for sequential excision, (4) a fluorescent marker and (5) a barcode DNA tag (Fig. 1b). Regulatory elements, expression cassettes and the barcode DNA tag were first individually mobilized into level 1 vectors using modular cloning principles (Supplementary Table 3)[34]. All 20 possible combinations of the four antibiotics and five fluorescent markers were assembled into a modular cloning-adapted pSEVA211-based backbone[35]. For each antibiotic–fluorescent marker combination, we constructed three to five ready-to-use vectors with distinguishable barcode DNA tags (Supplementary Table 5). Chromosomal integration into *attTn7* is enforced by the pSEVA211 vector backbone containing

the restrictive *R6K* origin of replication[36], which renders the vector unstable in most bacteria, and the negative selection marker *sacB* (Extended Data Fig. 2).

For profiling of microbial communities by amplicon sequencing, the barcode DNA tag is flanked by conserved bacterial *V5–V7 16S rRNA* sequences, conserved fungal *ITS1* and *ITS2* sequences, plant *ITS*-p4 and *ITS*-p5 sequences[37,38], and barcode-specific primer binding sites (Fig. 1b). Unique barcode DNA tags were generated by random blunt-end ligation of an equimolar ratio of ten different, double-stranded oligonucleotides (38 nucleotides (nts)), each consisting of four pyrosequencing-friendly barcodes, followed by DNA fragment size selection (Fig. 1c). The final barcode DNA tags preferentially consisted of an array of four oligonucleotides, resulting in an average length of approximately 150 nts per barcode (Fig. 1c). Preassembled barcode DNA tags were then integrated in exchange for a negative *ccdB* selection cassette previously integrated between primer binding sites of recipient vectors (Fig. 1d).

MoBacTag plasmid DNA was additionally used as spike-in DNA during library preparation for amplicon sequencing and comparative analysis (Fig. 1b and Supplementary Information). As the *16S*, fungal *ITS* and plant *ITS* primer binding sites flank the barcode DNA tag, read counts assigned to chromosomally integrated barcode DNA tags, bacterial *16S rRNA* or fungal *ITS* can be normalized to read counts specific to spike-in barcode DNA tags. These ratios can then be used to calculate bacterial-to-plant, fungal-to-plant or bacterial-to-fungal ratios. Finally, these ratios provide an estimate of the microbial load of the corresponding microbial kingdom in the sampled plant compartment (Figs. 1a and 2a).

### Validation of tags as spike and artificial *16S rRNA* sequence

Correlation of spike read counts with the spike DNA concentration was tested for each oligonucleotide pair in root and peat matrix samples to ensure that spike DNA abundance was reflected in read counts specific to barcode DNA tag (Fig. 2a–c and Extended Data Fig. 3a,b). We first normalized all read counts using spike 1 reads to compensate for different sequencing depths. Normalized spike-2-specific barcode read counts from amplicon sequencing with primers specific to *16S rRNA*, fungal *ITS*, plant *ITS* or the barcode correlated linearly over five orders of magnitude with spike 2 plasmid DNA concentrations in both root and peat matrix samples.

We compared the efficiencies of PCR amplification and DNA sequencing for natural *16S rRNA* with corresponding barcode DNA tags to test for previously reported PCR biases in microbiota profiling using *16S rRNA* sequences[39]. To exclude possible biases due to known variations in *16S rRNA* copy numbers in different bacterial taxa[40], different MoBacTags (termed tag119, tag120 and tag190) were inserted into wild-type *P. capeferrum* WCS358, and the WCS358:cyoB and WCS358:pqqF mini-*Tn5* transposon insertion mutants[29], at identical *Tn7* chromosomal integration sites. Short-read sequencing of individual MoBacTag-labelled bacteria revealed 3.5-fold higher read counts for tag119, 1.9-fold higher for tag120 and 1.8-fold higher for tag190 compared with the corresponding natural endogenous *16S rRNA* read counts of *P. capeferrum* WCS358 (Fig. 2d). These data indicate that despite identical chromosomal integration sites, primer binding sites and *16S rRNA* genetic backgrounds, individual barcode DNA tags exhibit different tag-to-*16S* read count ratios regardless of primers used for amplification (Fig. 2, Extended Data Fig. 3c,d and Supplementary Information). To next test whether the tag-specific amplification rates are unique to the *Tn7* locus, we also generated mutants of the root commensal Rhizobiales R13D by integrating MoBacTags (termed tag104, tag93, tag94, tag91) into different loci through homologous recombination. The data also showed tag-specific amplification with read counts that were 1.6-fold higher for tag104, 1.5-fold higher for tag93 and 1.4-fold higher for tag94 compared with the corresponding natural endogenous *16S rRNA* read counts (Fig. 2e). Fewer read counts were obtained for DNA tag91 than for endogenous *16S rRNA* (Fig. 2e),

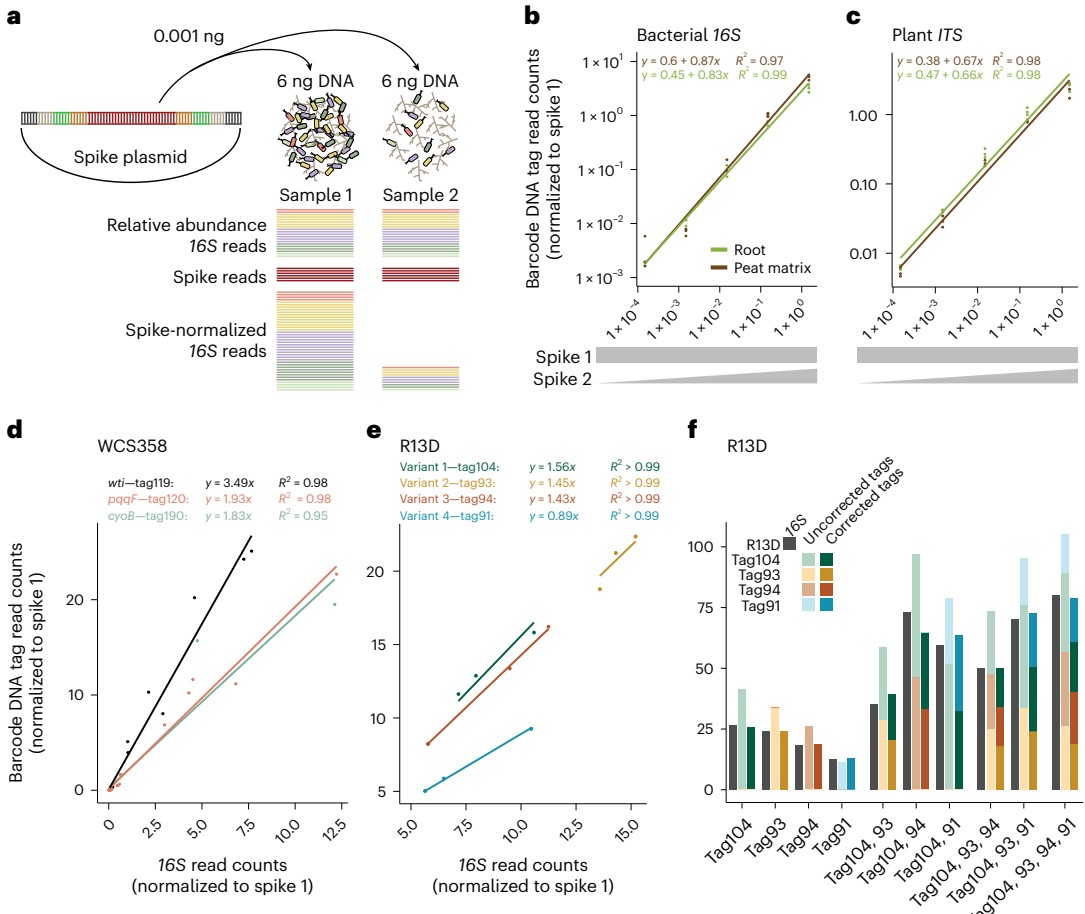

**Fig. 2 | Validation of the MoBacTag for spike normalization and strain abundance estimates during amplicon sequencing. a**, Principles of relative abundances and normalization using a spike plasmid in bacterial profiling. **b,c**, Linear correlation of normalized spike-specific read counts with spike concentrations obtained from amplicon sequencing of root ($n = 15$) and peat (peat) ($n = 15$) matrix samples using bacterial-*16S*-specific (799, 1,192; **b**) and plant-*ITS*-specific (*ITS*-p4, *ITS*-p5; **c**) primers. **d**, Linear correlation of normalized barcode-specific read counts with normalized *16S*-specific read counts for correction factor adjustment of individual chromosomally integrated barcodes

into WCS358 ($n = 16$ per variant, corresponding to $n = 10$ from root, $n = 3$ from peat matrix and $n = 3$ from input). **e**, Linear correlation of normalized barcode-specific read counts with normalized *16S*-specific read counts for correction factor adjustment of individual chromosomally integrated barcodes into Rhizobiales R13D ($n = 3$ per variant). **f**, Spike-normalized uncorrected and correction-factor-adjusted abundance of read counts for barcode and *16S* rRNA sequences within samples with increasing numbers of equimolar mixed Rhizobiales R13D barcode strains. The colours indicate the DNA barcode tags.

indicating tag-specific amplification bias for MoBacTags inserted at different, but also identical, chromosomal loci. To ensure that the read counts obtained from different barcode DNA tags accurately reflect the respective bacterial abundance, the tag-to-*16S* count ratios were used as a correction factor for the read counts specific to the barcode DNA tag. The corrected tag-specific read counts matched the read counts specific to *16S* rRNA obtained from the sequencing of pure bacterial cultures (Fig. 2f). Thus, correction factors need to be determined for each MoBacTag-labelled bacterial strain. We then investigated potential combinatorial effects of multiple MoBacTags within a sample. Genomic DNA of R13D strains carrying the four different barcode DNA tags were mixed in a 1:1 ratio with increasing complexity, that is, two, three or all four strains (Fig. 2f). Cumulative tag-specific read counts were comparable to *16S* rRNA read counts only after applying tag-specific correction factors irrespective of sample complexity, that is, two, three or four barcode DNA tags. Similar read counts were obtained for each tag per condition when using the tag-specific correction factors determined from the ratios of tag to *16S* reads. In summary, the number of MoBacTag-labelled strains within a condition does not change the number of tag-specific reads per strain, so the corrected tag-specific reads reflect strain abundance.

## MoBacTag labelling of diverse root microbiota members

As with any *Tn7*-mediated insertion, genome integration of the MoBac-Tag into the bacterial strain of interest requires four (not necessarily consecutive) working days, assuming that a DNA transformation proto-col has already been established for the bacterium of interest (Fig. 3a)[33]. For conjugation-based tagging of root-derived *Rhodanobacter* R179, *Tn7* attachment sites and natural antibiotic resistances were investigated. For conjugation, MoBacTag multigene vectors were mobilized into auxotrophic, conjugation-competent *Escherichia coli* BW29427 (ref. 41). In addition to selecting transformants based on MoBacTag-encoded antibiotic resistance, selection against antibiotic-resistant *E. coli* BW29427 was achieved by depletion of diaminopimelic acid (DAP), which is required for the survival of *E. coli* BW29427 but not the target strains[42]. The presence of the MoBacTag fragment was verified by PCR of bacterial colonies using plant-*ITS*-specific oligonucleotides (Fig. 3b; JT103 and JT108). The absence of PCR-amplified fragments from non-transformed bacterial colonies confirmed the specific binding of the plant *ITS* oligonucleotides to the MoBacTag in *Rhodanobacter* R179. In community profiling experiments with 15 bacterial strains selected from the *At*-R-SPHERE collection (synthetic communities (SynCom) modified from a previous study[43]), no bacteria-derived sequences were

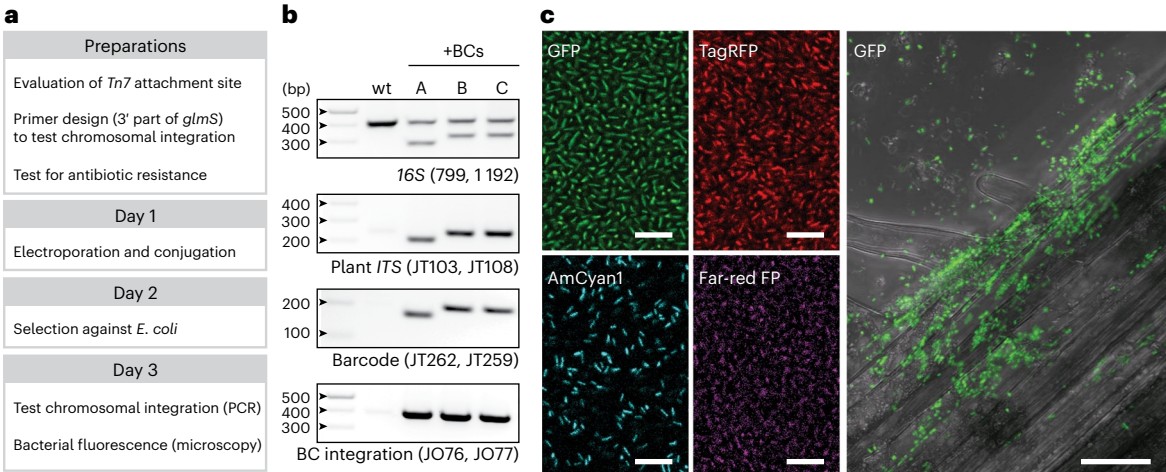

**Fig. 3 | Chromosomal integration of MoBacTags and expression of fluorescent tags. a**, General timeline for labelling a target strain by chromosomal MoBacTag integration. Depending on the target strain, *Tn7*-mediated insertion requires four (not necessarily consecutive) working days. **b**, Validation of MoBacTag transformation and chromosomal integration. DNA was extracted from transformed (+BC) and non-transformed *Rhodanobacter* R179 strains, PCRs were performed using indicated primers and amplicons were visualized on an agarose gel. To test barcode integration, *Tn7*-element- and *glmS*-specific primers

were used. This experiment was performed two times independently, and similar results were obtained. **c**, MoBacTag-labelled strains express fluorescent proteins. Expression of MoBacTag-encoded fluorescent protein was detected using live confocal laser scanning microscopy in liquid medium (four panels on the left; observed two times independently) and on *A. thaliana* roots (panel on the right; observed five times independently). Representative images are shown. Scale of bacteria in liquid culture (left), 10 μm; scale on roots (right), 20 μm.

recovered using the plant-*ITS*-specific oligonucleotides, supporting the target specificity of the plant *ITS* primers. In addition, transformants were genotyped with PCR using oligonucleotides specific to *16S rRNA*, resulting in two size-separable PCR products, as the barcode DNA tag is shorter than the endogenous *16S rRNA* amplicon (Fig. 3b; oligonucleotides 799 and 1192). Chromosomal integration was validated using a combination of *Tn7*-specific and *glmS*-strain-specific oligonucleotides (Fig. 3b)[33]. Further PacBio long-read genome re-sequencing of MoBacTag-labelled and wild-type *Arabidopsis*-root-derived Rhizobiales strains revealed no chromosomal rearrangements induced by the *Tn7*-mediated insertion (Extended Data Fig. 4a). Thus, the tagged strains differ from the wild type only by the chromosomally integrated MoBacTag, which could be validated by PCR-based genotyping. Comparisons of unlabelled and MoBacTag-labelled strains showed indistinguishable colony morphologies and growth curves in liquid medium (Extended Data Fig. 4b). Moreover, the composition of a microbial community was largely unaltered by the presence of a MoBacTag chromosomally integrated into the *P. capeferrum* WCS358 plant microbiota member (Extended Data Figs. 5 and 6 and Supplementary Information).

MoBacTags encoding different fluorescent markers were transformed into the root-derived commensal *Rhodanobacter* R179. Live cell imaging using confocal laser scanning microscopy was used to detect the expression of the four chromosomally encoded fluorescent markers driven by the *tac* promoter in bacteria cultivated on nutrient-rich medium or in association with the plant host (Fig. 3c). Expression of fluorescent proteins was also detected in MoBacTag-labelled *P. capeferrum* WCS358, *Pseudomonas simiae* WCS417, *Xanthomonas campestris* pv. *vesicatoria* and *Rhizobium* R13D, indicating a robust activity of the *tac* promoter in the corresponding bacterial taxa (Extended Data Fig. 4c).

### Spike normalization to determine microbial load in planta

To investigate whether the immunosuppressive activity of *P. capeferrum* WCS358 selectively promotes its own colonization or also influences root colonization by other members of the bacterial community, wild-type WCS358 and the immunosuppressive mutants WCS358:pqqF and WCS458:cyoB[29] were each, or in combination, co-inoculated with a taxonomically diverse SynCom consisting of 15 *Arabidopsis*-root-derived bacteria from the *At*-R-SPHERE culture

collection on germ-free *Arabidopsis* seedlings[43,44]. The corresponding wild-type genes contribute to the production of gluconic acid and its derivative 2-keto gluconic acid, which are proposed to suppress plant immunity by lowering extracellular pH locally[29]. To track individual strains in the SynCom, wild-type WCS358, WCS358:pqqF and WCS358:cyoB were labelled with MoBacTags differing in DNA barcodes, but an identical antibiotic resistance cassette, so that potential marginal effects of the MoBacTag on community establishment were identical for all conditions (Extended Data Fig. 5).

We first focused on the SynCom members by in silico depletion of WCS358 *16S rRNA* reads. Read counts relative to sample read depth indicated a higher relative abundance of *Streptomyces* spp. R1310 and a decreased relative abundance of *Microbacterium* spp. R61 in the presence of the WCS358:pqqF mutant compared with the WCS358 wild type, which is not indicative of altered absolute abundances (Fig. 4a). Normalization of *16S rRNA* reads to spike or plant-*ITS*-derived reads revealed a substantially reduced total bacterial load on roots colonized by the WCS358:pqqF mutant (Fig. 4b,c). Therefore, the altered relative abundances of *Streptomyces* spp. R1310 and *Microbacterium* spp. R61 in the presence of WCS358:pqqF might incorrectly hint at absolute changes due to the compositional nature of community profiling in the absence of normalization. As a result, further analyses were performed only on the spike-normalized data.

Constrained principal coordinate analysis of Bray–Curtis dissimilarities of spike-normalized *16S rRNA* reads showed that the reduced SynCom load on roots in the presence of the WCS358:pqqF mutant compared with wild-type WCS358 correlates with the formation of distinct communities (Fig. 4d,e). Seven SynCom members showed a significantly reduced root colonization capacity in the presence of WCS358:pqqF compared with the WCS358 wild type (Fig. 4f,g). The *pqqF* gene product is essential for the biosynthesis of pyrroloquinoline quinone (PQQ)[45], which serves as a redox-sensitive co-factor of several bacterial dehydrogenases including the glucose dehydrogenase required for the production of gluconic acid and its derivative[46]. The reduced total microbial load can be explained by a similar trend for half of all SynCom members, suggesting that PQQ biosynthesis and not gluconic acid production by wild-type WCS358 promotes root colonization by taxonomically diverse members of the root microbiota.

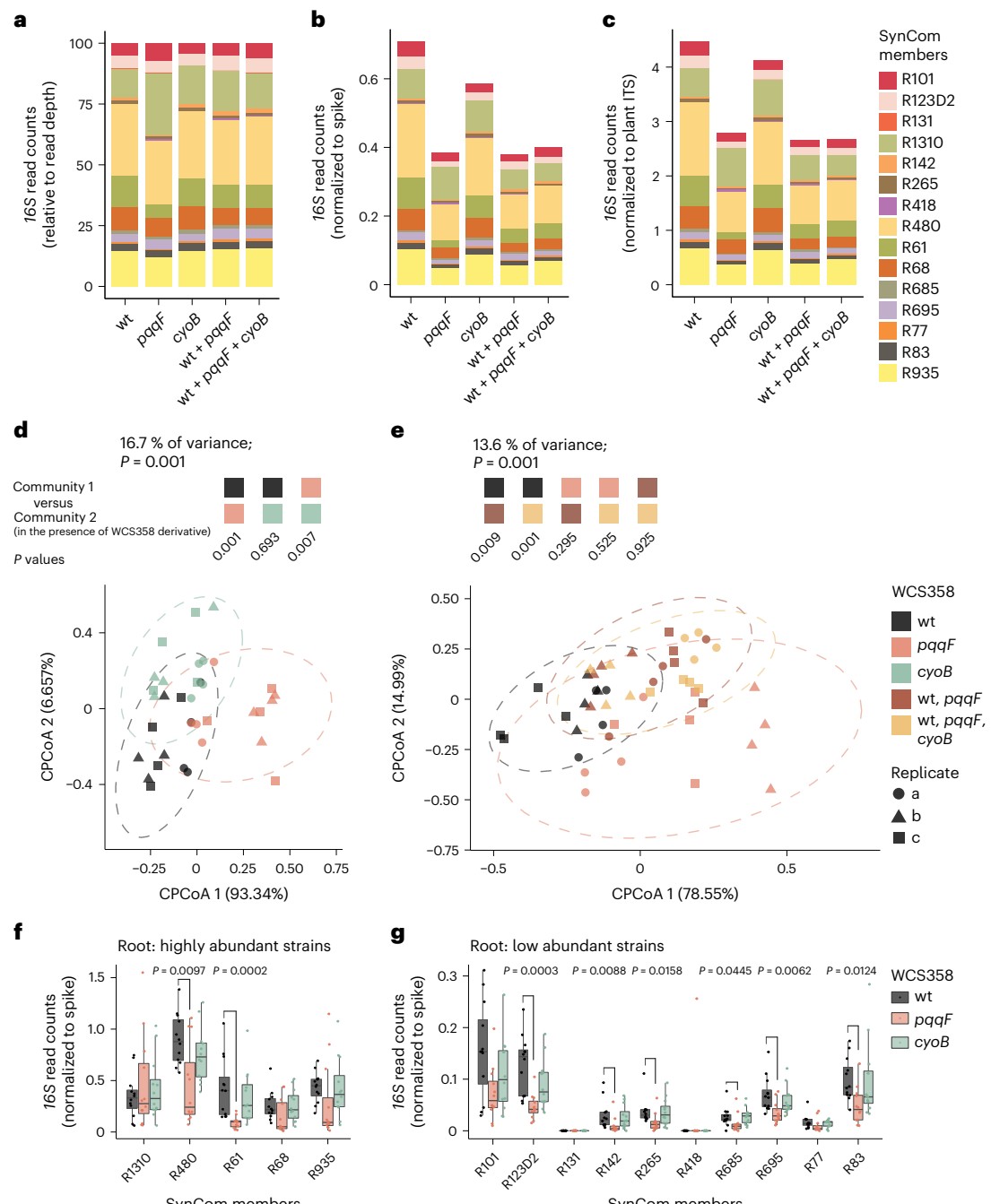

**Fig. 4 | *P. capeferrum* WCS358-encoded pqqF modulates the root microbiota.**
**a**–**c**, Relative (**a**), spike-normalized (**b**) and plant-*ITS*-normalized (**c**) abundance of SynCom members co-inoculated with *P. capeferrum* WCS358 wild type, and WCS358:pqqF or WCS358:cyoB mutant on *A. thaliana* roots, plotted as stacked bar plots. Colours indicate SynCom members. The mean of 12 biological samples collected from three independent replicates is shown. **d**,**e**, Constrained coordination of the microbial profile of a 15-member SynCom in the presence of either *P. capeferrum* WCS358 wild type (dark grey, *n* = 12), WCS358:pqqF (salmon, *n* = 12) or WCS358:cyoB (light green, *n* = 12) mutant (**d**), or WCS358 wild type plus WCS358:pqqF (brown, *n* = 12) or additional WCS358:cyoB (light orange, *n* = 12) on *A. thaliana* roots (**e**). The *n* values indicate biological samples collected from three independent replicates. The *P* values indicate statistical significance determined by permutational analysis of variance test between the compositions of SynCom 1 and SynCom 2 co-established with WCS358 derivatives indicated by the coloured boxes (permutation = 999). Ellipses correspond to Gaussian distributions fitted to each cluster (95% confidence interval). Shapes represent independent experiments. wt, wild type; CPCoA, constrained principal coordinate analysis. **f**,**g**, The normalized abundance of strains in the root compartment on co-inoculation with *P. capeferrum* WCS358 wild type (dark grey, *n* = 12), and the *pqqF* (salmon, *n* = 12) or *cyoB* (light green, *n* = 12) mutant: highly abundant (**f**) and low abundant (**g**) strains. The *n* values indicate biological samples collected from three independent replicates. The *P* values indicate statistical significance determined between the wild type and the mutants using a two-sided Dunn's test. The box plots centre on the median and extend to the 25th and 75th percentiles, and the whiskers extend to the furthest point within the 1.5× interquartile range.

Unexpectedly, however, root-associated communities established in the presence of wild-type WCS358 plus WCS358:pqqF showed only a slight shift towards communities containing wild-type WCS358 alone (Fig. 4b,c,e). Thus, the presence of the PQQ-deficient WCS358:pqqF strain has an unexpected dominant influence on root microbiota establishment that is not complemented by co-inoculation with the

PQQ-producing wild-type WCS358. This suggests that immunosuppression mediated by gluconic acid synthesis in wild-type WCS358 is insufficient to support the establishment of wild-type-like root communities in the presence of the WCS358:pqqF mutant. We hypothesize that this lack of *trans*-complementation by wild-type WCS358 might be related to PQQ consumption owing to proliferation of the WCS358:pqqF mutant, reducing the PQQ pool available for the SynCom. We tested whether WCS358 can import extracellular PQQ. Excess of D-glucose mediates growth inhibition through medium acidification by the activity of PQQ-dependent glucose dehydrogenase (GDH[46]). Consistent with this, the growth of the WCS358:pqqF mutant in vitro was not restricted by an excess of D-glucose compared with the growth of wild-type WCS358 (Extended Data Fig. 7). Chemical PQQ supplementation of the WCS358:pqqF mutant restored wild-type-like D-glucose-dependent growth restriction, suggesting that WCS358 imports PQQ.

Unlike WCS358:pqqF, the loss of *cyoB* in WCS358 had only a minor effect on the total bacterial load and the composition of the root-associated communities (Fig. 4d,f,g). As genetic depletion of *cyoB* in *P. capeferrum* WCS358 eliminates 2-keto-D-gluconic acid production but preserves residual gluconic acid biosynthesis in vitro[29], the wild-type-like community found here in the presence of WCS358:cyoB might be explained by residual gluconic acid production in planta.

### Tracking of near-isogenic strains in microbiota

Next, we asked whether the reduced root colonization by the WCS358:cyoB and WCS358:ppqF mutants reported from previous inoculation experiments in an unsterilized soil–sand matrix[29] can be reproduced in a peat-matrix-based gnotobiotic plant system[47] in the presence of a defined 15-member bacterial community. Consistent with the relative *16S rRNA* read counts for the 15 members of the SynCom (Fig. 4a), the interpretation of the relative *16S rRNA* read counts specific to *P. capeferrum* WCS358 was limited by the differential total microbial load between conditions (Fig. 5a). Accordingly, no difference in abundance was detected between the wild-type WCS358 and the WCS358:pqqF mutant when analysing the relative read counts. However, the spike-normalized abundances of *16S rRNA* reads from *P. capeferrum* WCS358 strains derived from either wild type, or WCS358:cyoB or WCS358:pqqF mutants, recapitulated the reduced root colonization by the mutants in our peat-based gnotobiotic plant system previously observed in an unsterilized soil–sand matrix (Fig. 5b). In agreement with published data, the abundance of wild-type WCS358 and WCS358 mutants in the peat matrix compartment was comparable (Fig. 5a,b)[29]. Thus, *pqqF* and *cyoB* are needed to specifically promote WCS358 colonization of the root compartment but are dispensable for bacterial growth in the peat matrix.

To test whether strain-resolved abundances of MoBacTags can be retrieved during community profiling of bacteria with identical *16S rRNA* sequences, the abundances of the MoBacTags incorporated into *P. capeferrum* WCS358 wild type, and the WCS358:pqqF and WCS358:cyoB mutants, were examined. Spike-normalized, corrected barcode DNA tag reads independently confirmed similarly reduced root and wild-type-like peat matrix colonization by the tested WCS358 mutants when only one WCS358 genotype was added to the 15-member SynCom at a time (Fig. 5c,d). Interestingly, co-inoculation with wild-type WCS358 did not elevate either WCS358:pqqF or WCS358:cyoB abundance in the root compartment (Fig. 5c). Thus, the WCS358 wild type did not *trans*-complement for the impaired root colonization of WCS358 mutants. On the contrary, in these mixed inoculation experiments (wild-type WCS358 plus WCS358 mutants), the abundance of wild-type WCS358 and to a greater extent that of WCS358:cyoB on roots decreased upon co-inoculation with WCS358:pqqF (Fig. 5c), as was the case for most other SynCom members (Fig. 4f,g). Furthermore, after co-inoculation of both WCS358 mutants, wild-type WCS358 and the 15-member SynCom, we detected fewer than 10 reads for WCS358:cyoB in all 12 root and most (8 out of 11) matrix samples. This corresponds

to a >100-fold reduction compared with wild-type WCS358 reads, indicating that WCS358:cyoB was essentially outcompeted in this condition (Fig. 5d). The tag-based strain abundances reveal previously unsuspected additional and distinct functions of *pqqF* compared with *cyoB* on roots. In summary, the MoBacTag allowed us to test for complementation or competition between strains encoding identical *16S rRNA* sequences during root microbiota establishment.

To show broad utility of MoBacTags, we inoculated MoBacTag-labelled strains *Rhizobium* R13C and R13D in two natural soils, each containing a highly complex resident microbial community, and in a Jiffy peat matrix (Extended Data Fig. 8a–d). Five weeks after cultivation of *Arabidopsis* Col-0 in these substrates, reads specific to the barcode DNA tags of R13C and R13D were detected in samples from unplanted control substrates, planted substrates and root compartments (Extended Data Fig. 8e–g). Thus, MoBacTag can also be used to track near-isogenic bacterial strains in highly complex resident communities of natural soils.

## Discussion

We have developed and validated DNA barcodes, which are co-amplified with natural endogenous bacterial *V5–V7 rRNA*, fungal *ITS* or plant *ITS* sequences by integrating respective primer binding sites into the DNA barcode. The MoBacTag enables direct tracking of multiple near-isogenic strains during conventional bacterial *16S rRNA* or fungal *ITS* community profiling. Since MoBacTag plasmids are based on modular cloning principles, each position within the multi-gene construct can be customized[34] by, for example, replacing *Tn7* elements with genomic regions for homologous recombination or inserting (1) additional community profiling primer binding sites into the DNA barcode, (2) an expression cassette for complementation approaches or (3) available modular cloning-compatible elements such as intensity-optimized fluorescent markers[48,49].

Bacterial DNA barcoding was applied to investigate antagonistic interactions of pseudomonad leaf strains co-occurring on *Arabidopsis*[28]. However, the DNA barcode architecture used does not allow co-amplification with endogenous marker genes, meaning that community analysis is restricted to barcoded strains only. The MoBacTag barcodes were semi-randomly assembled by random ligation of synthesized oligonucleotides consisting of 454 barcodes to avoid homopolymer or dipolymer tracts (Fig. 1). Unexpected PCR biases of individual 454-based barcodes required correction factors to determine the abundance of MoBacTag-labelled strains (Fig. 2d,e). As the error-corrected MoBacTags are chromosomally integrated as a single copy in most bacteria, tag-based abundance estimates can avoid biases due to variations in *16S rRNA* gene copy number from 1 to 15 between bacterial species[40,50]. Our spike-in plasmid architecture is similar to the architecture of other spike-in plasmids that require quantitative PCR measurements (Fig. 1a)[51,52]. Our spike plasmid allows for direct *16S rRNA* read count normalization and normalization to plant *ITS* read counts. The latter enables studying the composition and load of microbial communities in experiments with multiple plant species. However, differences in plant *ITS* copy number between plant species and accessions should be accounted for using correction factors. Besides spike-in normalization, host-associated microbe PCR (hamPCR)-based approaches are also used to measure microbial load and community composition[53]. MoBacTag and hamPCR approaches are complementary and could be combined to further increase the accuracy of host microbial load estimates.

The spike-in normalization revealed a significantly reduced SynCom load in the presence of the WCS358:pqqF mutant, but not WCS358:cyoB, which can be explained by significantly decreased abundances of 7 of the 15 SynCom members, each representing different core taxonomic lineages of the *A. thaliana* root microbiota (Fig. 4b,f,g)[43,44]. WCS358:pqqF and WCS358:cyoB are both impaired in the acidification of the extracellular space, although the WCS358:cyoB

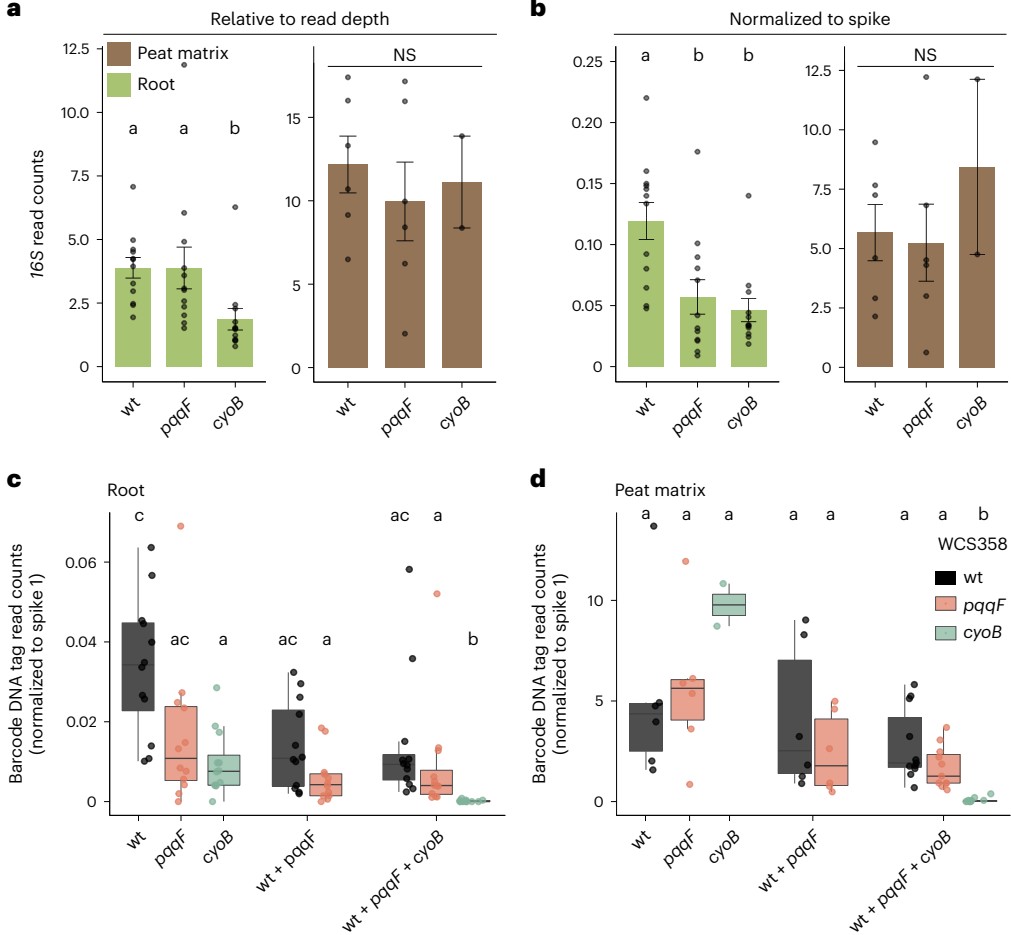

**Fig. 5 | MoBacTags recapitulate the root colonization deficit of WCS358 *pqqF* and *cyoB* mutants. a,b**, Relative (**a**) and spike-normalized (**b**) abundance of *P. capeferrum* WCS358 wild type, WCS358:cyoB and WCS358:pqqF in the root (*n* = 36) and peat matrix (*n* = 14) compartment quantified by *16S rRNA* read counts. The *n* values indicate biological samples collected from three independent replicates. The letters indicate statistical significance (*P* < 0.05) determined using a two-sided Dunn's test. The bar plots represent the mean ± s.e.m. **c,d**, The normalized abundance calculated using barcode-specific reads of *P. capeferrum* WCS358 wild type, and the *pqqF* or *cyoB* strains in the root (*n* = 60) (**c**) and peat matrix (*n* = 31) (**d**) compartment on co-inoculation with the 15-member SynCom alone, or in indicated mixtures. The *n* values indicate biological samples collected from three independent replicates. The letters indicate statistical significance (*P* < 0.05) determined using a two-sided Dunn's test. The box plots centre on the median and extend to the 25th and 75th percentiles, and the whiskers extend to the furthest point within the 1.5× interquartile range. NS, not significant.

mutant produces residual amounts of gluconic acid[29]. Despite shared functions in the biosynthesis of gluconic acid, the WCS358:pqqF mutant, and not the WCS358:cyoB mutant, reduced microbial load on roots, which is not *trans*-complemented by gluconic-acid-producing wild-type WCS358. Thus, the observed reduction in microbial load is unlikely to be a direct consequence of impaired gluconic-acid-mediated host immunosuppression. We propose instead that the co-factor PQQ serves as a common good within the root-associated bacterial community, as has been shown for other cofactors such as cobamides in bacterial co-cultures composed of different species[54]. Extracellular PQQ levels are probably relevant in bacterial communities, as PQQ import has been shown to be concentration dependent: the co-factor is imported by diffusion at high concentrations, whereas active TonB-dependent import is required at low concentrations, at least for *E. coli*[55]. Thus, the WCS358:pqqF mutant could reduce the extracellular PQQ pool on roots owing to lack of PQQ biosynthesis despite continuous PQQ consumption through WCS358:pqqF proliferation, resulting in PQQ deficiency in the bacterial community. This would also explain why WCS358:pqqF-mutant-specific community phenotypes are not *trans*-complemented by wild-type *P. capeferrum* WCS358 in planta.

MoBacTag DNA barcodes generated in vitro were designed for tracking near-isogenic bacterial strains in community contexts across multiple generations. These strains cannot be distinguished by endogenous barcodes such as the *V5–V7 16S rRNA* region. The presence of the dominant-negative *ccdB* selection marker in the recipient vector (Fig. 1) should enable use of MoBacTags for pool transformation approaches, each with randomly loaded DNA barcode tags (Fig. 1)[17]. This opens up future opportunities for MoBacTag-based lineage tracking during experimental evolution of microbial communities[56], as potential compensatory growth responses of community members to changes in labelled lineages can be tracked in parallel. Addition of primer binding sites for other eukaryotic hosts will increase the versatility of MoBacTag vectors for studies of any host–microbiota interaction.

## Methods

### Bacterial media and growth conditions

Level 1 vectors were transformed into *E. coli* DH5 and DB3.1 for *ccdB*-encoding plasmids (Supplementary Table 1). Level 2 pBCC and pBC vectors were transformed into *E. coli* DB3.1λpir[57] and BW29427, also known as WM3064[41], respectively (Supplementary Table 1). pTNS3 (ref. 58) was also cloned into *E. coli* BW29427. Transformation of the *R6K* origin of replication encoded on MoBacTags into bacteria with methyl-specific restriction systems was enabled by chromosomal integration of *pir2*, which is responsible for R6K-based replication, into a non-methylating

conjugation-competent *E. coli* (ET12567/pUZ8002)[59,60]. Therefore, *pir2* was transformed into *E. coli* ET12567/pUZ8002 via *Tn7*-mediated chromosomal integration similar to the procedure of MoBacTag labelling. *pir2* was first mobilized into pUC18-miniTn7-Gent-GW[61] using Gateway cloning according to the user manual.

*E. coli* were cultivated in Luria–Bertani medium (25 g l⁻¹ Luria–Bertani medium; Sigma) at 37 °C, and commensal bacterial strains were cultivated in 0.5 tryptic soy broth (TSB, 15 g l⁻¹; Sigma) or TY medium (5 g l⁻¹ tryptone, 3 g l⁻¹ yeast extract, 10 mM $CaCl_2$) at 25 °C. Media were supplemented with 15 g l⁻¹ Bacto Agar (Difco) for solidification. Antibiotics or DAP were added to the media, when necessary, at the following concentrations: streptomycin (Sm, 100 μg ml⁻¹), spectinomycin (Sp, 100 μg ml⁻¹), tetracycline (Tc, 10 μg ml⁻¹), gentamicin (Gm, 25 μg ml⁻¹), kanamycin (Kn, 50 μg ml⁻¹) and DAP (50 μg ml⁻¹).

### Assembly of unique DNA barcodes by random ligation of oligonucleotides

Complementary oligonucleotides (Supplementary Table 2) were mixed at an equimolar ratio (10 μM), incubated at 94 °C for 2 min and gradually cooled. Double-stranded oligonucleotides were then mixed in a ratio of 1:1 and prepared for blunt-end ligation using the End-It DNA End-Repair Kit according to the user manual (Biosearch Technologies). Ligation was performed overnight at 4 °C using T4 DNA ligase (New England Biolabs) followed by heat inactivation for 10 min at 80 °C. The preassembled barcodes were finally selected for 200–300 bp fragments by BluePippin from Sage Science at the Max Planck Genome Centre, Cologne, Germany (https://mpgc.mpipz.mpg.de/home/). Size-selected arrays of ligated oligonucleotides were then cloned between primer binding sites of pBCC vectors (see next paragraph).

### Generation of MoBacTag plasmids using the modular cloning strategy

Expression cassettes, terminator sequences and mini-*Tn7* elements were amplified using PrimerStar Max DNA Polymerase from Takara (templates indicated in Supplementary Table 3). PCR fragments were cloned into level 1 recipient plasmids (Supplementary Table 3) from the MoClo Toolkit by a restriction and ligation reaction with BsaI and T4 DNA ligase (New England Biolabs) using ligase buffer[34,62]. BsaI, BpiI and SmaI recognition sites were removed by altering a single nucleotide, without changing the encoded amino acid, in the recognition site using primers with restriction-site-mutating sequences (Supplementary Tables 2 and 3). The *ccdB* cassette was flanked with SmaI recognition sites and amplicon sequencing primer binding sites by consecutive PCRs with tailed primers (Supplementary Table 2) and then cloned into the level 1 recipient plasmid plCH47761. The antibiotic markers (pBCC030, pBCC031, pBCC032) were flanked with Flp recombinase target (FRT) sites by two consecutive PCRs using FRT site-encoding tailed primers, whereas for pBCC033, the FRT sites were already present in the donor plasmid (Supplementary Table 3). First, a green fluorescent protein (GFP)-encoding plasmid was assembled (pBCC029) by amplifying GFP from pUC18-mini-Tn7T-Tp-gfpmut3 and pTAC from pLM449 with tailed primers integrating BsaI restriction sites, and this was cloned into the plCH47751 level 1 receptor by a restriction and ligation reaction with BsaI and T4 DNA ligase using ligase buffer. Finally, the pTAC promoter and coding sequence of the fluorescent markers were amplified from pLM426 derivatives (Supplementary Table 3) and combined with the pBCC0029 recipient plasmid including a terminator sequence by In-Fusion HD from Takara. All plasmids for this study were purified using the NucleoSpin Plasmid kit (Macherey-Nagel). Level 1 domesticated sequences were then assembled into a pSEVA211-based[35] level 2 recipient vector by a BpiI restriction and ligation reaction, resulting in BarCode Construction (pBCC) multi-gene constructs (Supplementary Table 3). To this end, a red fluorescent protein (tagRFP) expression cassette was amplified by PCR integrating BpiI recognition sites and cloned into the pSEVA211 by a restriction and ligation reaction with BsaI,

EcoRI, HindIII and T4 DNA ligase (New England Biolabs). Preassembled barcodes were inserted by SmaI restriction and ligation as follows: 45 cycles of 5 min at 16 °C, and 5 min at 30 °C followed by 5 min 95 °C. Then, fresh SmaI was added and incubated at 30 °C for 30 min resulting in final MoBacTag vectors. Barcode tag sequences were identified for individual *E. coli* BW29427 colonies from pooled transformation with MoBacTag vectors by colony PCR with primers MobacTag barcode F/R (Supplementary Table 5), followed by Sanger sequencing (Eurofins Scientific). Sequencing results were analysed in CLC Main Workbench (QIAGEN). The plasmids generated for the MoBacTag kit will be made available via Addgene.

### Labelling of bacterial strains with MoBacTags using the mini-*Tn7* system

MoBacTag vectors were transferred into commensal bacterial strains (Supplementary Table 1) by triparental mating. Saturated liquid cultures of the recipient commensal strain, *E. coli* strain BW29427/pTNS3 and BW29427/pBC were mixed in a 1:2:2 ratio and incubated for 24–48 h at 25 °C. Afterwards, transformants were selected on 0.5 TSB or TY medium containing 10% sucrose and the corresponding antibiotics. Bacterial DNA was extracted by re-suspending a bacterial colony in 25 μl buffer I (25 mM NaOH, 0.2 mM EDTA, pH 12) followed by incubation at 95 °C for 30 min and addition of 25 μl buffer II (40 mM Tris–HCl). The genomic insertion of the MoBacTag was validated by PCR as described in a previous study[33].

### *AttTn7* box conservation across the *At*-R-SPHERE culture collection

The *glmS* sequences were extracted from the whole-genome assemblies of every strain included in the At-SHPERE culture collection[44]. The last 12 amino acids or the last 36 DNA bases from the extracted *glmS* sequences were aligned and visualized with the software WebLogo (https://weblogo.berkeley.edu/logo.cgi).

### Bacterial genome assembly

The Max Planck Genome Centre, Cologne, Germany (https://mpgc.mpipz.mpg.de/home/), performed the DNA isolation of wild-type and MoBacTag-labelled strains and also the sequencing on the Pacific Biosciences Sequel IIe platform. Reads were assembled de novo using the Hifiasm software (https://github.com/chhylp123/hifiasm). To compare wild-type and MoBacTag-labelled strains, we used Mauve software for reordering the contigs (https://darlinglab.org/mauve). Genomes were compared by generating a dot plot on genome scale using the Genome Pair Rapid Dotter (gepard) software (https://doi.org/10.1093/bioinformatics/btm039).

### MoBacTag-labelled and unlabelled *Rhodanobacter* growth in liquid monoculture

MoBacTag-labelled and unlabelled *Rhodanobacter* R179 strains were grown in six replicates each in 0.5 TSB as individual cultures in a 96-well bacterial culture plate (Greiner-CELLSTAR 96-well plate, transparent, flat bottom; Sigma-Aldrich) at 25 °C. Absorbance was measured every 10 min, 10 s after 20 s of shaking (290 rpm), using a microplate reader (Infinite M200 PRO, Tecan) at 600 nm. The mean values of four measurements per well were used for the analyses.

### PQQ- and glucose-dependent *P. capeferrum* growth in liquid monoculture

*P. capeferrum* wild-type WCS358 and the WCS358:pqqF mutant were grown in four replicates each in modified XVM2 minimal medium (20 mM NaCl, 10 mM $(NH4)_2SO_4$, 5 mM $MgSO_4$, 1 mM $CaCl_2$, 0.01 mM $FeSO_4$, 10 mM succinate, 0.03% Casamino acids) supplemented with 110 mM D-glucose and 3 μM PQQ as individual cultures in a 96-well bacterial culture plate (Greiner-CELLSTAR 96-well plate, transparent, flat bottom; Sigma-Aldrich) at 25 °C. Absorbance was measured every

10 min, 10 s after 20 s of shaking (290 rpm), using a microplate reader (Infinite M200 PRO, Tecan) at 600 nm. The mean values of four measurements per well were used for the analyses.

## Visualization of fluorescently labelled commensal bacteria

MoBacTag-labelled bacterial strains were harvested from 0.5 TSB plates and re-suspended in 10 mM $MgSO_4$. The root colonization assay was performed as follows: surface-sterilized *A. thaliana* Col-0 seeds were sown on agar plates (1% Bacto agar, BD Biosciences) prepared with MS/2 (as described for the Gnoptopot system) and supplemented with MoBacTag-labelled *Rhodanobacter* R179 or Rhizobiales R13D at a final concentration of $OD_{600} = 0.0005$ (ref. 63). Washed bacteria ('Bacterial microbiota reconstitution experiment' in Methods) were mixed into the medium before solidification. After 14 days of growth under short-day conditions (10 h light, 21 °C; 14 h dark, 19 °C), bacteria were visualized on roots. Confocal laser scanning microscopy was performed on a Zeiss LSM880 inverted confocal scanning microscope. Pictures were taken with an LD C-Apochromat 40×/1.1 water immersion objective. To image root colonization, Z-stacks were generated, and maximum intensity projections were compiled. The following excitation and detection windows were used: GFP: 488 nm, 493–598 nm; cyan fluorescent protein: 458 nm, 472–528 nm; tagRFP: 561 nm, 582–754 nm; and far-red fluorescent protein: 561 nm, 591–759 nm.

## Investigation of MoBacTag amplification biases by qPCR

For qPCR, genomic DNA extracted from WCS358, WCS358:pqqF and WCS358:cyoB as described in 'Labelling of bacterial strains with MoBac-Tags using Mini-Tn7 system' in Methods was used. Dilution series were used for qPCR using the iQ SYBR Green Supermix (Bio-Rad). Reactions (20 µl) were prepared with 3 µl genomic DNA, 10 µl SYBR Green Supermix and 0.4 µl of each primer (10 µM, Supplementary Table 6). qPCR was performed using the CFX ConnectTM Real-Time System (Bio-Rad) with the following conditions: 95 °C for 3 min; 95 °C for 15 s, 65 °C for 15 s and 72 °C for 15 s for 4 cycles; and 95 °C for 15 s, 57 °C for 15 s and 72 °C for 15 s for 39 cycles followed by a melting curve.

## Bacterial microbiota reconstitution experiments

Saturated bacterial liquid cultures were pelleted by centrifugation at 8,000*g* for 5 min, followed by two washes with 10 mM $MgSO_4$. Equivalent amounts of each strain were combined to yield the desired SynComs with an optical density ($OD_{600}$) of 2. Aliquots of individual strains and the SynComs were taken and stored at −80 °C. The inoculum solution was prepared with MS/2 (2.22 g l⁻¹ Murashige and Skoog basal salts, Sigma; 0.5 g l⁻¹ MES anhydrous, BioChemica; adjusted to pH 5.7 with KOH) and the SynCom at a final $OD_{600}$ of 0.02. We used the Gnotopot system[47] to grow *A. thaliana* Col-0 plants with the bacterial SynComs (Supplementary Table 6). Each pot was inoculated with the bacterial SynCom by decanting 10 ml of the inoculum solution. With the use of a syringe, the excess liquid was removed from the box. *A. thaliana* seeds were surface sterilized by incubation in 70% ethanol twice for 5 min, followed by a brief wash with 100% ethanol. Seeds were then washed three times with sterile water and cold stratified for 2 days. Six sterilized seeds were placed on the matrix of each pot (Jiffy-7 pellets, Jiffy Products, https://www.jiffygroup.com/) and incubated under short-day conditions for 5 weeks (10 h light, 21 °C; 14 h dark, 19 °C). Roots were harvested by thoroughly removing attached soil using sterile water and forceps. Root and peat matrix samples were collected in Lysing Matrix E tubes (FastDNA Spin Kit for Soil, MP Biomedicals) and frozen in liquid nitrogen. Samples were stored at −80 °C until DNA isolation, which was performed using the FastDNA Spin Kit for Soil according to the user manual (MP Biomedicals).

## Cultivation of MoBacTag-labelled *Rhizobium* strains in soil and Jiffy peat substrates

The MoBacTag-labelled *Rhizobium* R13C and R13D inoculum was prepared as above and inoculated in native Cas[64] and Golm[64] soil and non-autoclaved Jiffy peat (Jiffy-7 pellets, Jiffy Products, https://www.jiffygroup.com/) substrates to a final concentration ($OD_{600} = 0.2$ per strain). Eight surface-sterilized *A. thaliana* Col-0 seeds were sown on top and cultivated under short-day conditions for 5 weeks in a greenhouse (10 h light, 21 °C; 14 h dark, 19 °C). Pots were watered using deionized tap water throughout the experiment. Harvesting was performed as described for 'Bacterial microbiota reconstitution experiments' in Methods.

## Bacterial community profiling by amplicon sequencing

Library preparation for Illumina MiSeq sequencing was performed as described previously[5], except for the addition of 0.001 ng of pBCC023 or pBCC084 plasmid DNA per reaction to the master mix of PCR1, as spike. The final ratio of sample (6 ng) to spike (0.001 ng) DNA per PCR1 is 6,000. The oligonucleotides used for amplicon sequencing of *16S rRNA*, fungal *ITS* and plant *ITS* sequences are listed in Supplementary Table 5. In all experiments, multiplexing of samples was performed by single or double indexing (only forward barcoded oligonucleotides for single indexing or forward and reverse barcoded oligonucleotides for double indexing). The indexed amplicon libraries were pooled, purified using Ampure (Beckman Coulter) and sequenced on the Illumina MiSeq platform.

To validate MoBacTag as a spike-in plasmid using the bacterial-*16S*-, fungal-*ITS*-, plant-*ITS*- and barcode-specific primers, 0.15 ng of pBCC069 plasmid was mixed with 5 ng of extracted DNA from root and peat matrix samples, prepared by Getzke et al.[65]. The standard curve was prepared with pBCC084 plasmid in tenfold dilution series from 1.5 ng to 0.00015 ng per PCR.

## Processing of gene amplicon data

Amplicon sequencing data from SynCom experiments were de-multiplexed according to their barcode sequences and quality filtered using the USEARCH (v.10.0.240) and QIIME (v.2 2021.2) pipeline[66]. Paired-end reads were merged using the flash2 (v.2.2.00) software[67]. Quality-filtered merged paired-end reads were then aligned to reference amplicon sequences using Rbec (v.1.8.0)[11]. For the plant *ITS* sequencing data, only single-end reads were processed, because a sequence length of pITS (742 bp) cannot be merged from the Illumina 2 × 300 bp sequencing run. The reference sequences were extracted from whole-genome assemblies of every strain included in the SynCom, from the MoBacTag barcodes and from whole-genome assembly of *A. thaliana* Col-0 (TAIR9 assembly, www.arabidopsis.org). We checked that the fraction of unmapped reads did not substantially differ between compartments and experiments. Count tables were generated and used for downstream analyses of diversity in R (v.4.0.0) using the R package vegan[68]. Amplicon data were visualized using the ggplot2 R package[69].

## Normalized quantification of amplicon sequencing

Amplicon reads assigned to the spike were used to normalize plant *ITS* and bacterial *16S rRNA* read counts similar to what was done in a previous study[51]. Identical amounts of spike-in plasmid DNA included for plant *ITS* and *16S rRNA* library preparation were used for cross-normalization using the following equations:

$$\text{Normalized bacterial reads (Nb)} = \frac{\text{Reads, bacteria (Rb)}}{\text{Reads, spike (Rs}_{16S})}$$

$$\text{Normalized plant reads (Np)} = \frac{\text{Reads, plant (Rp)}}{\text{Reads, spike (Rs}_{pITS})}$$

$$\text{Normalized bacterial reads by plant reads} = \frac{\text{Rb} \times \text{Rs}_{pITS}}{\text{Rp} \times \text{Rs}_{16S}} = \frac{\text{Nb}}{\text{Np}}$$

## Statistics and reproducibility

All experiments were performed in three full-factorial (biological and technical) replicates. Bacterial abundances were compared using an ANOVA test, followed by Tukey's post hoc test ($\alpha = 0.05$). Statistical tests on beta-diversity analyses were performed using a permutational analysis of variance test with 999 random permutations. Whenever box plots were used in figures, data were represented as median values (horizontal line), $Q1 - 1.5 \times$ interquartile range (boxes) and $Q3 + 1.5 \times$ interquartile range (whiskers).

## Reporting summary

Further information on research design is available in the Nature Portfolio Reporting Summary linked to this article.

## Data availability

Raw amplicon reads and genome assemblies have been deposited in the European Nucleotide Archive under the accession number PRJEB61076. For the *A. thaliana* Col-0 sequence, the TAIR9 assembly was used (www.arabidopsis.org). The bacterial *At*-R-SPHERE 16S sequences were obtained from the website https://www.at-sphere.com. Source data are provided with this paper.

## Code availability

The scripts used for the computational analyses described in this study are available at https://github.com/thouinjulien/MoBacTag, to ensure replicability and reproducibility of these results.

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

## Acknowledgements

We thank D. Becker and E. Logemann for technical support. We acknowledge P. Poole (University of Oxford, UK) for providing the pLM449 plasmid containing the tac promoter and F. Getzke (Max Planck Institute for Plant Breeding Research, Germany) for providing root and peat matrix samples containing fungal DNA. We thank N. Donnelly, J. Stuttmann and R. Berendsen for reading and editing the paper. Funding was provided by the Max Planck Society and the German Research Foundation (DFG) under the German Excellence Strategy, EXC number 2048/1 project 390686111 for R.G.-O. and P.S.-L., and SPP 2125 DECRyPT for K.-W.M. and P.S.-L. J.T. was supported by the Alexander von Humboldt Foundation.

## Author contributions

P.S.-L., J.O., K.-W.M., R.G.-O. and B.H. conceptualized the methodology. J.O. and J.T. designed the experiments. R.T.N. cloned coding sequences for fluorophores. J.O. and J.T. generated MoBacTag plasmids. J.O. assembled barcode DNA tags and tested their diversity. J.T. tested spike-in normalization and amplification biases by multiple MoBacTag-tagged strains. J.O. performed microscopy and the proof-of-principle experiment. J.O., J.T. and P.Z. analysed the data. J.O. and J.T. produced the figures. J.O. and P.S.-L. wrote the paper with contributions from all co-authors.

## Funding

## Competing interests

The authors declare no competing interests.

# Resource

## Additional information

**Extended data** is available for this paper at

**Supplementary information** The online version
contains supplementary material available at

**Correspondence and requests for materials** should be addressed to
Paul Schulze-Lefert.

**Peer review information** *Nature Microbiology* thanks
David Johnston-Monje, Derek Lundberg and the other, anonymous,
reviewer(s) for their contribution to the peer review of this work.
Peer reviewer reports are available.

**Publisher's note** Springer Nature remains neutral with regard to
jurisdictional claims in published maps and institutional affiliations.

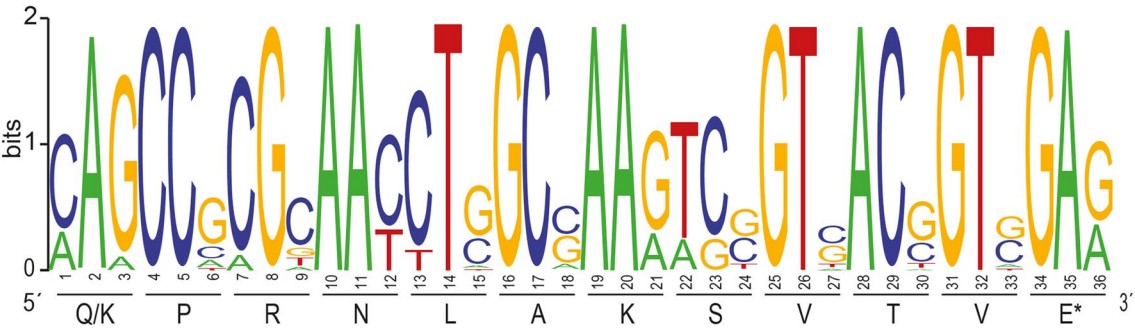

**Extended Data Fig. 1 | Conservation of the *Tn7* binding site in the *At*-SPHERE culture collection.** *Tn7* transposase binding site within the last coding 36 nts of the *glucosamine-6-phosphate synthetase* (*glmS*) gene, displayed as a DNA sequence logo for all *At*-SPHERE genome drafts. The letter size indicates the nucleotide abundance at each position. The encoded amino acids are depicted below.

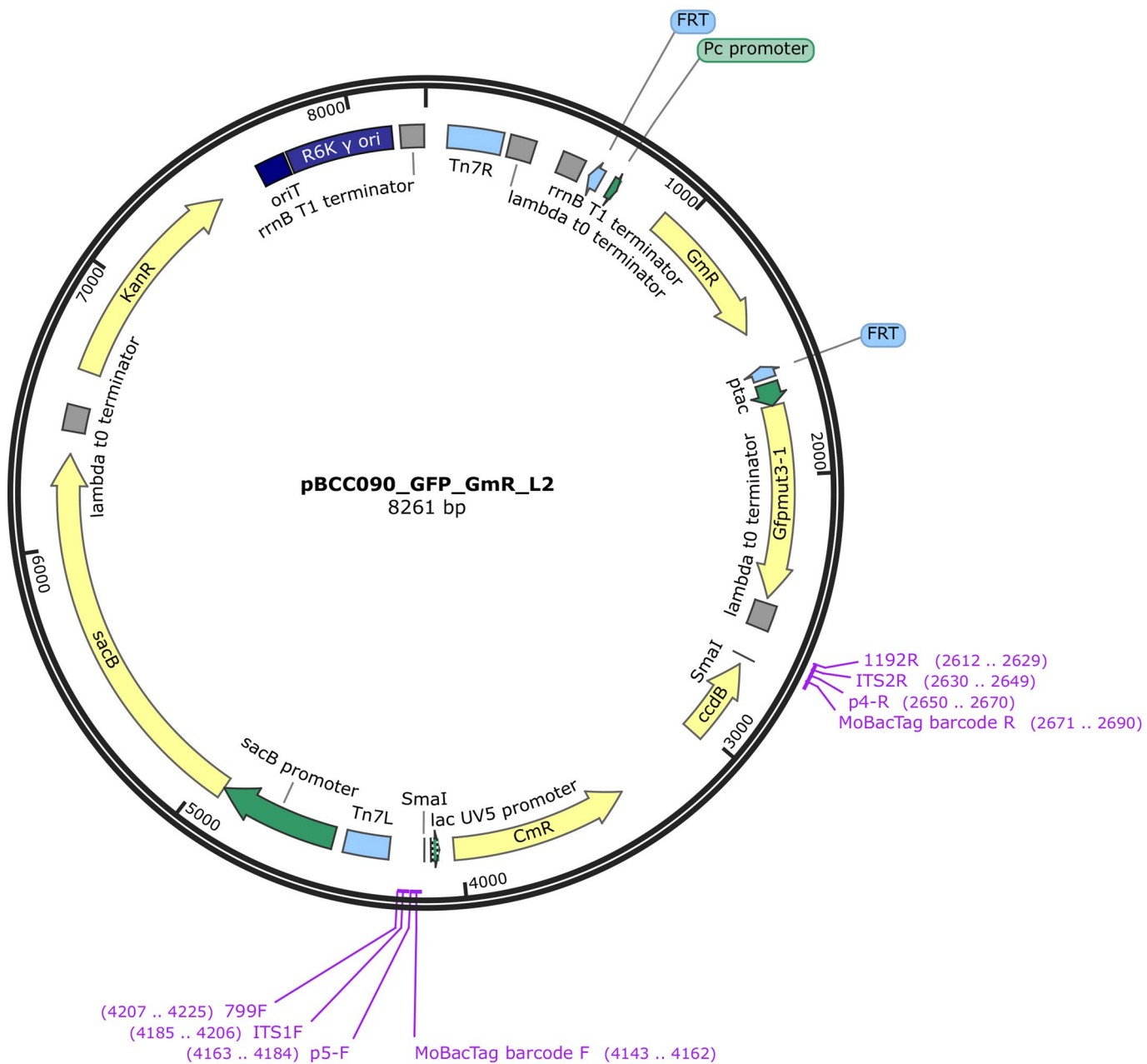

**Extended Data Fig. 2 | Plasmid map of an *N-acetyltransferase aac* and *GFP*-encoding pBCC recipient plasmid.** Plasmid map, created with SnapGene, indicating coding sequences (yellow), promoter sequences (green), terminator sequences (grey), origin of transfer and replication (dark blue), protein binding sites (light blue), primer binding sites (purple) and *Sma*I restriction sites.

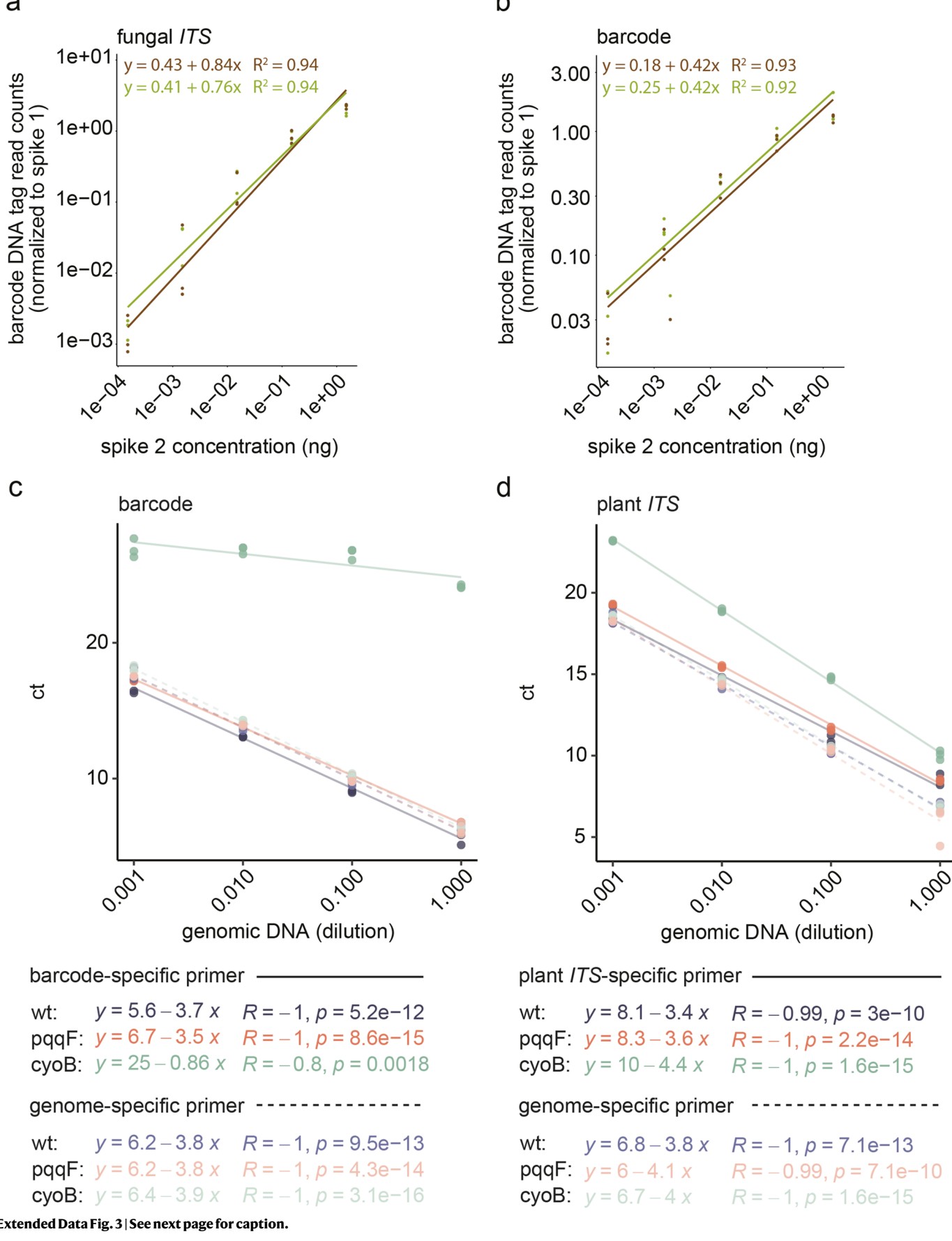

**Extended Data Fig. 3 | See next page for caption.**

**Extended Data Fig. 3 | Extended validation of the MoBacTag for spike normalization and barcode DNA tag-specific amplification efficiencies.**
**a,b**, Linear correlation of normalized spike-specific read counts with spike concentrations obtained from amplicon sequencing of root (n = 15) and peat (n = 15) matrix samples using fungal *ITS*- (ITS1/ITS2; a) and barcode-(JT259/JT262; b) specific primers. **c,d**, Amplification efficiencies of different barcode DNA tags determined by qPCR using dilution series of genomic DNA from MoBacTag-labeled wild-type WCS358, WCS358:pqqF and WCS358:cyoB using barcode-specific (c), plant *ITS*-specific (d) and inter-gene chromosome-specific primers. R values indicate Pearson correlation coefficients and *p*-values indicate significance by a two-sided t-test.

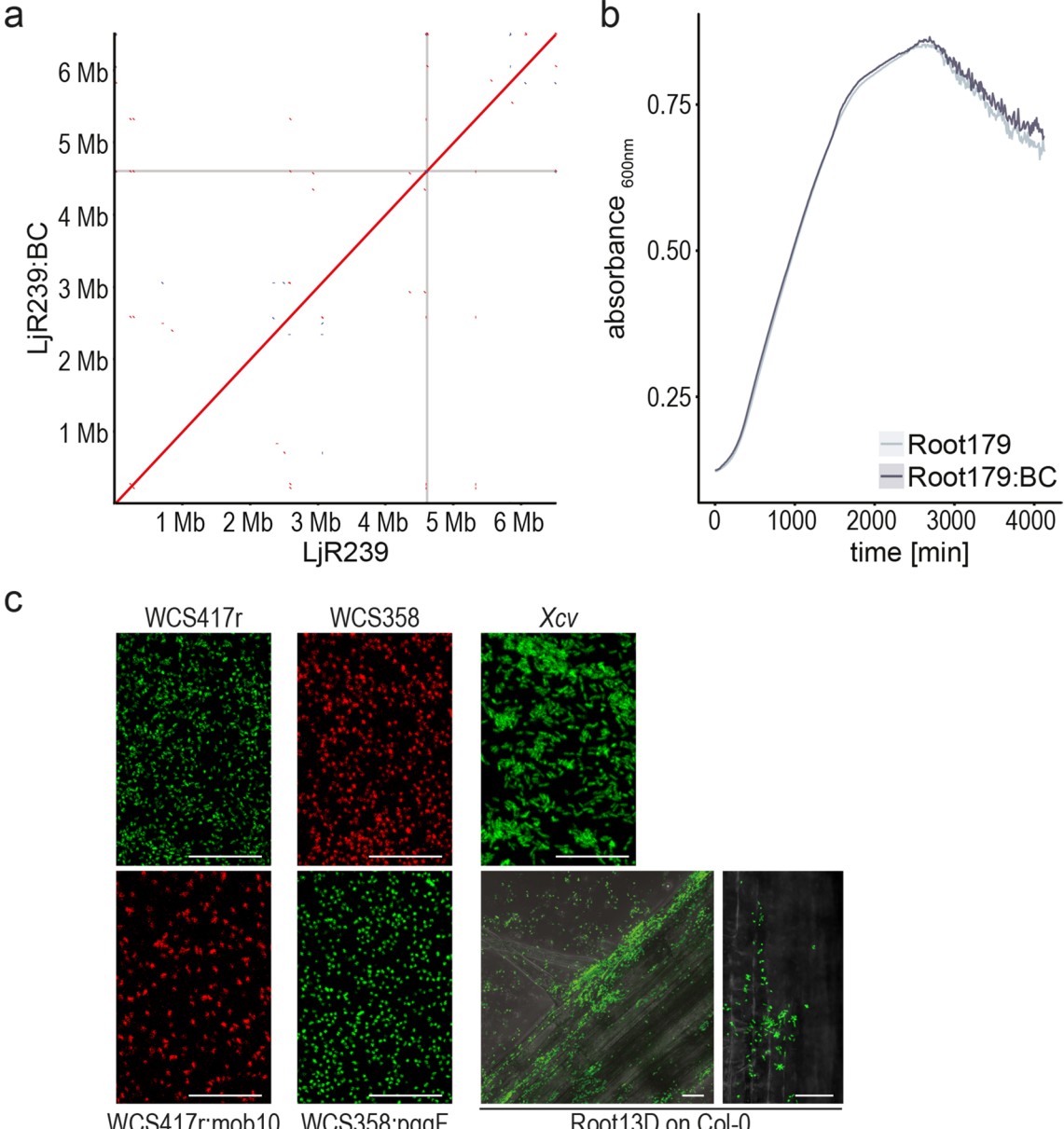

**Extended Data Fig. 4 | Chromosome integrity and expression of fluorescent tags in taxonomically distinct bacteria. a**, Alignments of PacBio genome assemblies from the unlabeled and MoBacTag-labeled Rhizobiales LjR239 isolated from *Lotus japonicus* roots. MoBacTag insertion site is indicated by grey lines. **b**, Bacterial growth of MoBacTag-labeled or unlabeled *Rhodanobacter* R179 in liquid rich (0.5 TSB) medium in monocultures indicated by absorbance

($\lambda$ = 600 nm). **c**, Expression of fluorescent proteins from chromosomally-integrated MoBacTags. Expression of MoBacTag-encoded fluorescent protein was detected using live confocal laser scanning microscopy in liquid medium (observed two times independently) and on *A. thaliana* roots (observed three times independently). Representative images are shown. Xcv: *Xanthomonas campestris* pv. *vesicatoria*. Scale: 20 μm.

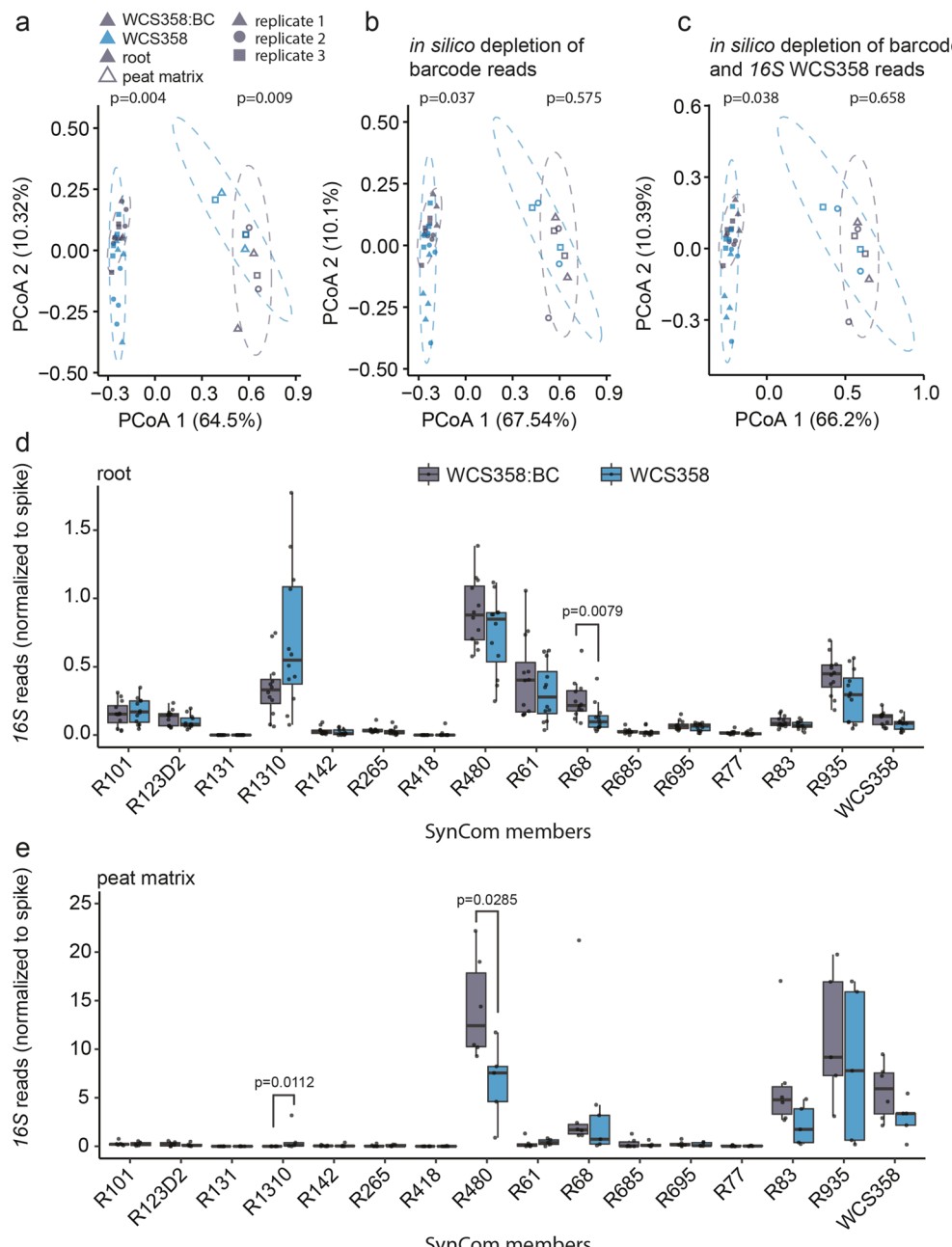

**Extended Data Fig. 5 | The influence of MoBacTag on root microbiota establishment. a-c**, Coordination of the spike-normalized reads from a 15-member synthetic community with wild-type or MoBacTag-labeled *P. capeferrum* WCS358 in the *A. thaliana* root (n = 24) and peat matrix (n = 10) compartment (a), upon *in silico* depletion of barcode reads (b) and additional *in silico* depletion of WCS358 *16S rRNA* reads (c). Shapes represent the compartment and colors represent WCS358 derivatives. n values indicate biological samples collected from three independent replicates. Ellipses correspond to Gaussian distributions fitted to each cluster (95% confidence interval). P values indicate statistical significance determined using a permutational analysis of variance (PERMANOVA) test between communities including the labeled or unlabeled WCS358 derivate (permutation = 999, P < 0.05). **d,e**, The normalized abundance of individual strains in the root (n = 24) (d) and peat matrix (n = 10) (e) compartment upon co-inoculation with the labeled (WCS358:BC) or unlabeled (WCS358) strain. n values indicate biological samples collected from three independent replicates. P-values indicate statistical significance determined using two-sided Dunn's test. The box plots center on the median and extend to the 25th and 75th percentiles, and the whiskers extend to the furthest point within the 1.5x interquartile range.

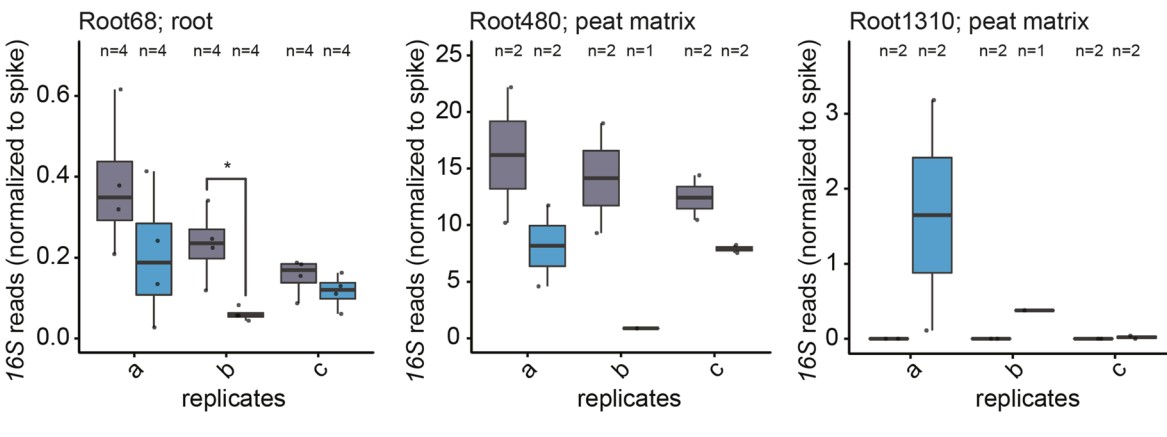

**Extended Data Fig. 6 | The influence of a MoBacTag on selected SynCom members in independent experiments.** The normalized abundance of strains that had significantly different abundances after co-inoculation with the labeled (WCS358:BC) or unlabeled (WCS358) strain when all replicates are analyzed individually (Extended Data Fig. 5d, e) for each independent experiment. n values indicate biological samples. The box plots center on the median and extend to the 25th and 75th percentiles, and the whiskers extend to the furthest point within the 1.5x interquartile range.

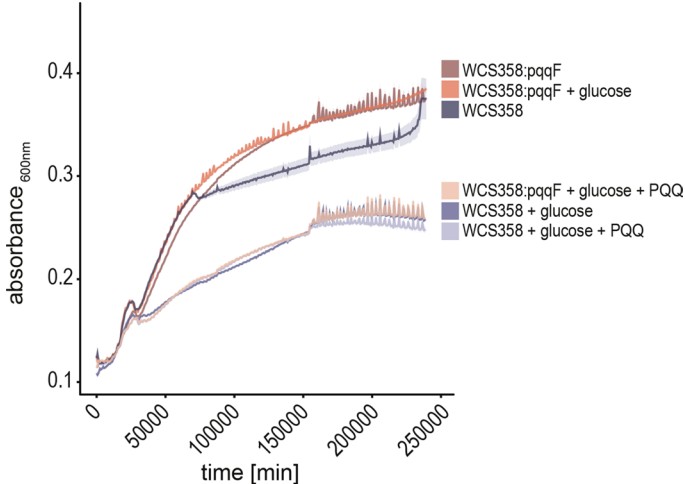

**Extended Data Fig. 7 | Chemical complementation of WCS358:pqqF mutant by extracellular PQQ.** Bacterial growth of *Pseudomonas capeferrum* wild-type WCS358 and the WCS358:pqqF mutant in unbuffered modified XVM2 minimal medium supplemented with 110 mM glucose and 3 μM PQQ in monocultures indicated by absorbance (λ = 600 nm).

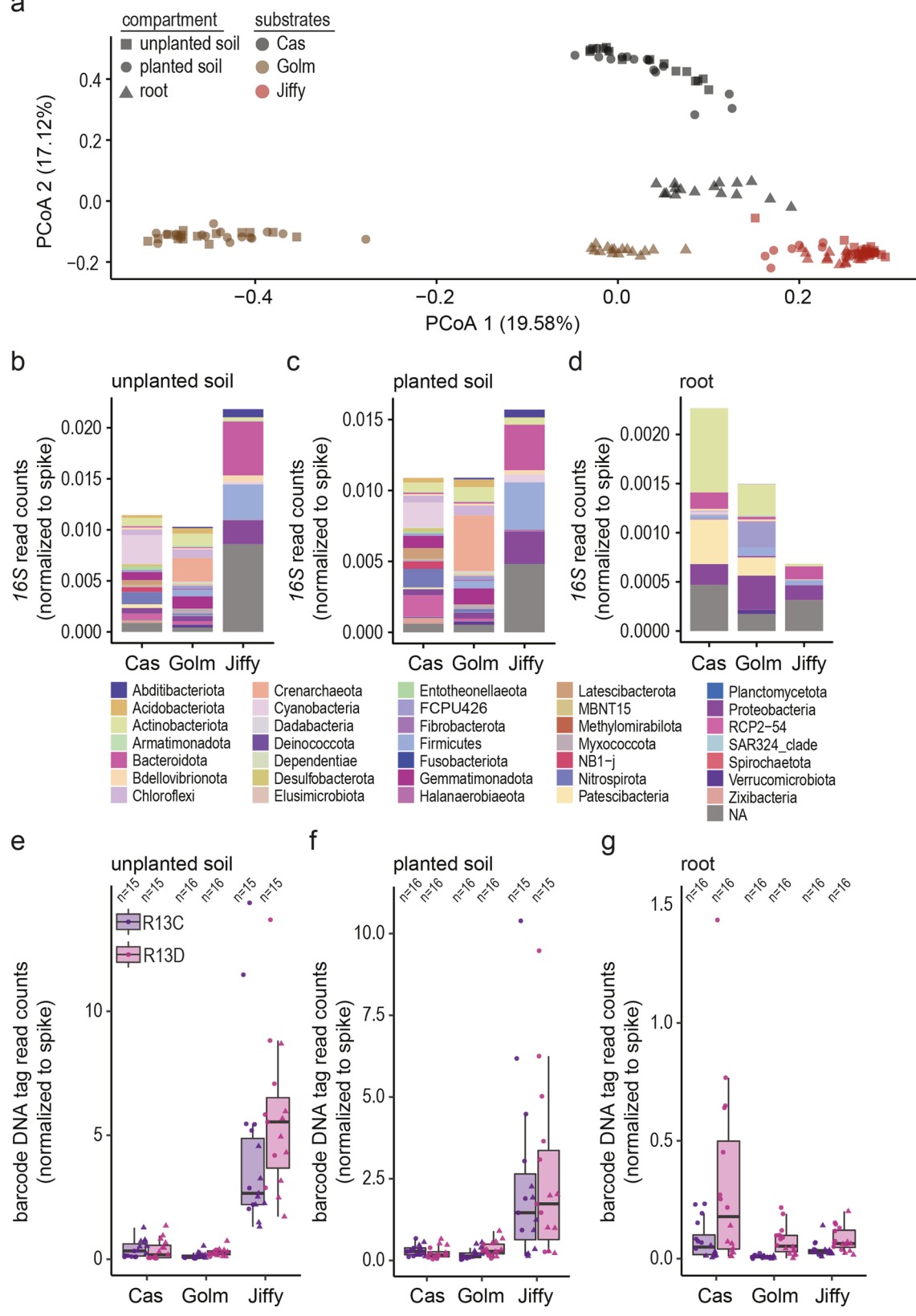

**Extended Data Fig. 8 | See next page for caption.**

**Extended Data Fig. 8 | Detection of barcode DNA tag-specific reads from MoBacTag-labeled *Rhizobium* R13C and R13D in resident microbial communities of two soil types and a peat substrate. a**, Unconstrained PCoA plot showing differences between bacterial communities in unplanted and planted Cologne agricultural soil (Cas[1]; black; n = 16, n = 16), soil collected near Golm[1] (brown; n = 16, n = 16), Jiffy peat matrix (https://www.jiffygroup.com/; red; n = 16, n = 15), and Arabidopsis roots five weeks after plant cultivation in the respective substrates (n = 16 for each substrate). **b-d**, Spike-normalized *16S rRNA* reads at phylum level of unplanted (b), planted (c) Cas or Golm soils or Jiffy peat

matrix, and Arabidopsis Col-0 roots. For this analysis, *Rhizobium* R13C- and R13D-specific *16S rRNA*- and barcode DNA tag-specific reads were depleted *in silico*. **e-g**, Normalized barcode-specific read counts of *Rhizobium* R13C (purple) and R13D (pink) in the indicated unplanted substrates (e), planted substrates (f), and root compartment (g). n values indicate biological samples collected from two independent replicates. The box plots center on the median and extend to the 25th and 75th percentiles, and the whiskers extend to the furthest point within the 1.5x interquartile range.

# Reporting Summary

## Statistics

For all statistical analyses, confirm that the following items are present in the figure legend, table legend, main text, or Methods section.

| n/a | Confirmed | |
|---|---|---|
| ☐ | ☒ | The exact sample size (*n*) for each experimental group/condition, given as a discrete number and unit of measurement |
| ☐ | ☒ | A statement on whether measurements were taken from distinct samples or whether the same sample was measured repeatedly |
| ☐ | ☒ | The statistical test(s) used AND whether they are one- or two-sided *Only common tests should be described solely by name; describe more complex techniques in the Methods section.* |
| ☐ | ☒ | A description of all covariates tested |
| ☐ | ☒ | A description of any assumptions or corrections, such as tests of normality and adjustment for multiple comparisons |
| ☐ | ☒ | A full description of the statistical parameters including central tendency (e.g. means) or other basic estimates (e.g. regression coefficient) AND variation (e.g. standard deviation) or associated estimates of uncertainty (e.g. confidence intervals) |
| ☐ | ☒ | For null hypothesis testing, the test statistic (e.g. *F*, *t*, *r*) with confidence intervals, effect sizes, degrees of freedom and *P* value noted *Give P values as exact values whenever suitable.* |
| ☒ | ☐ | For Bayesian analysis, information on the choice of priors and Markov chain Monte Carlo settings |
| ☐ | ☒ | For hierarchical and complex designs, identification of the appropriate level for tests and full reporting of outcomes |
| ☒ | ☐ | Estimates of effect sizes (e.g. Cohen's *d*, Pearson's *r*), indicating how they were calculated |

*Our web collection on statistics for biologists contains articles on many of the points above.*

## Software and code

Policy information about availability of computer code

Data collection | 16S amplicon data were collected using a MiSeq Illumina sequencer. Reads were demultiplexed with QIIME, merged with FLASH2, cleaned with USEARCH and the amplicon sequencing variant table was generated with Rbec. The genomes were sequenced with a Sequel IIe Pacific Biosciences sequencer and assembled with Hifiasm.

Data analysis | QIIME v2 2021.2
FLASH2 V.2.2.00
USEARCH v10.0.240
Rbec v1.8.0
R v4.0.0
R package vegan v2.6-4
R package ggplot2 v3.4.4
Hifiasm v0.16.1
Mauve v2015-02-25
gepard v2.1.0
WebLogo
Costumized code is available here: https://github.com/thouinjulien/MoBacTag

For manuscripts utilizing custom algorithms or software that are central to the research but not yet described in published literature, software must be made available to editors and reviewers. We strongly encourage code deposition in a community repository (e.g. GitHub). See the Nature Portfolio guidelines for submitting code & software for further information.

## Data

Policy information about availability of data

All manuscripts must include a data availability statement. This statement should provide the following information, where applicable:
- Accession codes, unique identifiers, or web links for publicly available datasets
- A description of any restrictions on data availability
- For clinical datasets or third party data, please ensure that the statement adheres to our policy

> Raw 16S rRNA amplicon and genome reads are deposited in the European Nucleotide Archive (ENA) under the accession number PRJEB61076. For the A. thaliana Col-0 sequence the TAIR9 assembly was used (www.arabidopsis.org). Bacterial At-R-SPHERE 16S sequences were retrieved from the website https://www.at-sphere.com.

## Research involving human participants, their data, or biological material

Policy information about studies with human participants or human data. See also policy information about sex, gender (identity/presentation), and sexual orientation and race, ethnicity and racism.

| | |
|---|---|
| Reporting on sex and gender | NA |
| Reporting on race, ethnicity, or other socially relevant groupings | NA |
| Population characteristics | NA |
| Recruitment | NA |
| Ethics oversight | NA |

Note that full information on the approval of the study protocol must also be provided in the manuscript.

# Field-specific reporting

Please select the one below that is the best fit for your research. If you are not sure, read the appropriate sections before making your selection.

☒ Life sciences ☐ Behavioural & social sciences ☐ Ecological, evolutionary & environmental sciences

For a reference copy of the document with all sections, see nature.com/documents/nr-reporting-summary-flat.pdf

# Life sciences study design

All studies must disclose on these points even when the disclosure is negative.

| | |
|---|---|
| Sample size | Previous experiments using the same gnotobiotic system were used as reference to determine sample size and to allow confident statistical analyses (e.g. Wippel et al., 2021. Nature Microbiology; Kremer et al., 2021. Nature Protocols). For each figure, the sample size is indicated as "n" in the corresponding figure legend. No statistical methods were used to determine sample size. |
| Data exclusions | No data were excluded from the analyses. |
| Replication | The gnotobiotic experiment (Figure 4 to 6) was performed with 3 independent preparations of the corresponding SynComs (biological replicates) and with at least 8 technical replicates of planted pots for each independent SynCom preparations.<br>The molecular biology to validate the oligonucleotides from Figure 2 was performed with at least 15 technical replicates.<br>The inoculation of MoBacTag-labeled Rhizobia strains into native soils was performed with 2 independent experiments and with 8 technical replicates per experiment per soil type. No attempt of replication was excluded from the analysis. |
| Randomization | Position of boxes with planted pots was randomized in the growth chamber during the course of the experiments. |
| Blinding | The samples were harvested and processed by multiple researchers. Each SynCom condition or compartment were processed by every researcher.<br>After setting-up the experiment, boxes and samples were labeled with unique numbers to blind the researcher during the harvesting and processing. |

# Reporting for specific materials, systems and methods

We require information from authors about some types of materials, experimental systems and methods used in many studies. Here, indicate whether each material, system or method listed is relevant to your study. If you are not sure if a list item applies to your research, read the appropriate section before selecting a response.

## Materials & experimental systems

| n/a | Involved in the study |
|-----|----------------------|
| ☒ ☐ | Antibodies |
| ☒ ☐ | Eukaryotic cell lines |
| ☒ ☐ | Palaeontology and archaeology |
| ☒ ☐ | Animals and other organisms |
| ☒ ☐ | Clinical data |
| ☒ ☐ | Dual use research of concern |
| ☐ ☒ | Plants |

## Methods

| n/a | Involved in the study |
|-----|----------------------|
| ☒ ☐ | ChIP-seq |
| ☒ ☐ | Flow cytometry |
| ☒ ☐ | MRI-based neuroimaging |

