## [Peer Review File · Nature Microbiology]

Peer Review Information

Journal: Nature Microbiology

Manuscript Title: Chromosomal barcodes for simultaneous tracking of near-isogenic bacterial strains in plant microbiota

Corresponding author name(s): Professor Paul Schulze-Lefert

Reviewer Comments & Decisions:

Decision Letter, initial version:

Message: 13th June 2023

Dear Professor Schulze-Lefert,

Thank you for your patience while your manuscript "Simultaneous tracking of near-isogenic bacterial strains in synthetic *Arabidopsis* microbiota by chromosomally-integrated barcodes" was under peer-review at Nature Microbiology. It has now been seen by 3 referees, whose expertise and comments you will find at the end of this email.

Although they find your work of some potential interest, they have raised a number of concerns that must be addressed before we can return any revised version to reviewers. In particular, as pointed out clearly by reviewer 1, there needs to be experimental evidence of the broad utility of your method. This is the most important concern raised in my view, because IF the method can be shown to be broadly applicable, the usefulness is proven. Whilst a very well worked out application or biological finding is a good addition to any paper, my view is that the broad utility is more important, so whilst the application requires some revisions, it is the range of strains that is the major concern. If you do not want to demonstrate broad utility, I can engage my colleagues at Nature Communications to see if they are willing to overrule this concern.

Should further experimental data allow you to address these criticisms, we would be happy to look at a revised manuscript.

Please include a data availability statement as a separate section after Methods but before references, under the heading "Data Availability". This section should inform readers about the availability of the data used to support the conclusions of your study. This information includes accession codes to public repositories (data banks for protein, DNA or RNA sequences, microarray, proteomics data etc...), references to source data published alongside the paper, unique identifiers such as URLs to data repository entries, or data set DOIs, and any other statement about data availability. At a minimum, you should include the following statement: "The data that support the findings of this study are available from the corresponding author upon request", mentioning any restrictions on availability. If DOIs are provided, we also

2strongly encourage including these in the Reference list (authors, title, publisher (repository name), identifier, year). For more guidance on how to write this section please see: <http://www.nature.com/authors/policies/data/data-availability-statements-data-citations.pdf>

- * Include a "Response to referees" document detailing, point-by-point, how you addressed each referee comment. If no action was taken to address a point, you must provide a compelling argument. This response will be sent back to the referees along with the revised manuscript.
- * If you have not done so already we suggest that you begin to revise your manuscript so that it conforms to our Resource format instructions at <http://www.nature.com/nmicrobiol/info/final-submission>. Refer also to any guidelines provided in this letter.
- * Include a revised version of any required reporting checklist. It will be available to referees (and, potentially, statisticians) to aid in their evaluation if the manuscript goes back for peer review. A revised checklist is essential for re-review of the paper.

Note: This url links to your confidential homepage and associated information about manuscripts you may have submitted or be reviewing for us. If you wish to forward this e-mail to co-authors, please delete this link to your homepage first.

Nature Microbiology is committed to improving transparency in authorship. As part of our efforts in this direction, we are now requesting that all authors identified as 'corresponding author' on published papers create and link their Open Researcher and Contributor Identifier (ORCID) with their account on the Manuscript Tracking System (MTS), prior to acceptance. This applies to primary research papers only. ORCID helps the scientific community achieve unambiguous attribution of all scholarly contributions. You can create and link your ORCID from the home page of the MTS by clicking on 'Modify my Springer Nature account'. For more information please visit www.springernature.com/orcid.

If you wish to submit a suitably revised manuscript we would hope to receive it within 6

months. If you cannot send it within this time, please let us know. We will be happy to consider your revision, even if a similar study has been accepted for publication at Nature Microbiology or published elsewhere (up to a maximum of 6 months).

Yours sincerely,

Reviewer Expertise:

Referee #1: Plant Microbiome

Referee #2: Plant-microbe interactions, computational approaches.

Referee #3: Plant microbiome, microbial ecology

Reviewer Comments:

Reviewer #1 (Remarks to the Author):

This paper capitalizes on the widely-used Tn7 transposon in order to create a system to track closely-related bacterial strains in a community, and reveals some of the insights possible with such a system using gluconic acid mutants in *Pseudomonas*. The Tn7 transposon has an advantage over other transposons in that it has a predictable insertion site in those bacterial genomes in which it can integrate. The idea of using artificial genetic barcodes to distinguish closely related individuals is not novel, and the authors cite appropriate literature. What is novel is the particular design of the construct that contains the barcode, the design of the barcode itself, and the attention to robust validation of the results. Overall the paper is nicely written.

>>>>Major concerns

There are two major questions I have after reading this. The first is, can it work in my bacteria? The authors looked in their culture collection bioinformatically and found theoretical Tn7 insertion sites (glmS) in 98.6% of their genomes. Then in the paper, they use the Tn7 tags in *Pseudomonas* only. Other published papers using Tn7 for broad host range transformation attempt more strains <https://www.frontiersin.org/articles/10.3389/fmicb.2018.03052/full> and struggle with some of them. In this regard, it's a bit of an empty claim that all these genomes have insertion sites if it's not clear insertion will work in them. I would have really appreciated an estimate of the fraction of plant-associated bacterial genomes that can be transformed and transposed using standard techniques.

The other major question was related to the example case of the gluconic acid mutants. The authors presented intriguing results and mentioned that the ppqF mutant that could not make gluconic acid had a dominant phenotype in the bacterial community when mixed with wildtype *Pseudomonas*. They wrote "the immunosuppression mediated by gluconic acid synthesis in wild-type WCS358 is insufficient to support the establishment of wild type-like root communities in the presence of the WCS358:ppqF mutant." But then, they do not explain why. Is it simply that the ppqF mutant competes for resources with the wild-type and then the wild-type isn't abundant enough, and then for this reason the gluconic acid isn't abundant enough? Also the cyoB mutant the authors mention had a mild effect compared to ppqF, and the

authors suggest it could be (line 363) the “residual gluconic acid production in planta”, but then no followup. These were unsatisfying conclusions to the main biological message in this paper. Perhaps because this paper focuses on methods this could be said to be outside the scope of this work, but then this makes the first major concern above more relevant.

One major question I had with the methods regards the differences in abundance for different MoBacTags in the same genetic background. The authors tested three tags in *Pseudomonas* (lines 2019 to 216). These data were collected using the 16S primers, which allowed also comparing tag abundance to 16S abundance. However, it is unclear from this whether the same tag-specific correction factors would also be needed if using the tag-specific primers instead of the 16S primers, for instance, or another primer set. Is there an intrinsic difference in the ability of each barcode to be amplified in PCR, or does it depend on what PCR primers are used?

>>>> Minor concerns

Line 206-210 “To exclude possible...”

> It seems the authors are saying that they take three clones of WCS358 and put a different tag in the Tn7 site in each one. However, due to the wording about “the same 16S rRNA genetic background” etc, this is confusing. It can be worded more simply.

>Line 218 “homologous recombination”

Why use homologous recombination? Did Tn7 integration fail here?

Line 219 “Tag104”

> This tag is mentioned but not defined until the end of the sentence where it says “(termed tag104, tag93...” Best would be to say “We also generated and tested four variants of the root commensal Rhizobiales Root13D with MoBacTags (tag 104, tag93, and 94)” with the tags defined right after they are mentioned.

Line 221 “DNA tag91”

> What is tag91 in? It wasn't referred to before.

Line 223 “at identical and different chromosomal integration sites”

> How can something be identical and different? Please rewrite.

Line 233-244 about tag correction factors..

> Interesting that each tag needs a correction factor. Can the authors speculate in the discussion why this may be the case?

Line 288

> Strain Root418 was less than 4% so the authors deleted it from analysis. Why? What's the harm to leave it? How is this justified? How do the authors know it was irrelevant?

Line 296 “which suggests..”

>It has been known for a long time that roots affect the microbes and have different communities than soil. So the use of the word “suggests” is not appropriate, because it looks like this is new information.

Line 308 to 313:

> Says a few syncom members differed depending on treatment, but the results weren't consistent. This paragraph makes one worried there is a problem, and starts giving details of individual strains that are affected, and then says the problem isn't so bad and the results

aren't significant. Perhaps it can be simplified to just make the main point that differences observed were not significant.

>FRT sites should work in theory, but not demonstrated. Would be nice to include guidance, at least, how to take advantage of them.

Line 331 to 337

>Can phrase in a way that shows the authors know how to interpret relative abundance. By saying "read counts... appeared to indicate enhanced colonization of Streptomyces.. " it seems the authors are admitting they got fooled, but it would be better to just say "read counts indicated a higher relative abundance of Streptomyces... which should not be misinterpreted as higher absolute abundance... Normalization of ..."

Line 336 "altered relative abundances... are likely an artefact"

> Relative abundances are just math.. they aren't an artefact. Could say the relative abundances might incorrectly imply a change to absolute abundance or something like that.

Line 396 "essentially outcompeted"

> Can the authors use a number? What does this qualifier "essentially" actually mean?

Line 434-436

> I know how a spike in works, but I am confused how it can "eliminate potential 454 barcode- and thereby sample-specific PCR bias".

Line 442 "single copy"

> In some genomes, there is more than one integration site, so this is not strictly true

Line 456-459

> This sentence was confusing.. please rewrite or split into two sentences.

Line 487 "tag-to-plant ITS"

>What drove the choice of a multiple copy number gene like plant ITS that differs between accessions vs. a single-copy plant gene for plant normalization?

>>>>FIGURES

1B

>Why is the synthetic DNA tag green and red, and everything else is just 1 color?

2A

>Put the color legend in Figure 2A rather than 2D.. I couldn't find it at first

4 "Ellipses correspond to"

>There are no ellipses... a-c were not especially clear to me, and I wasn't sure exactly how to interpret, but perhaps the ellipses would help with that.

5 B and C

> This was a clear and consistent result showing importance of normalizing to the plant!

5D and E

> The colored boxes at the top showing condition 1 vs condition 2.. confusing meaning.. how do the colors relate to other things.. and "conditions" are not mentioned in the legend, so it's hard to figure it out..it seems I compare condition 1 to condition 2 directly below and get the

p-value shown? Please clarify in legend.

Reviewer #2 (Remarks to the Author):

Ordon, Thouin et al. propose a new technology, modular bacterial tags (MoBacTags), which can be used to quantitatively characterize the relative abundances of various bacteria at the resolution of the subspecies/strain. The quantitation ability of MoBacTags are calibrated and characterized, and are finally used in a proof-of-principle experiment with two *Pseudomonas* mutant strains of the same species (i.e. identical 16s sequences) which have known phenotypes. The design of the MoBacTag vector is smart as it combined multiple combinations of antibiotic marker genes, multiple combinations of fluorescent markers, and synthetic DNA tag that is flanked by multiple spikes which enable many types of data normalization. The authors invested thought and lab work in calibration to convince the reader that the method is fairly reliable. The checking of the method was thorough. Most of the paper is dedicated to the establishment of this method, and in this sense it is a bit too technical in our view, for a method that is interesting and useful but overall we think will not be a game changer in the plant microbiome field. A unique usage which the method enables is the mixture of multiple near-isogenic strains in the same experiment as was shown for example in the wt+pqf experiments. The effect of pqf deletion on the quantity of many members of the community is impressive to observe but for this result any spike that allows accurate quantification (and not just report a change in community composition as usually done in microbiome studies) can be used and the barcode was not critical if I understand it correctly.

Major revisions

1. Figure 1A is critical for understanding the idea behind the method. Currently it is not clear enough. For example: it seems that "black DNA" refers to different things in every illustration (specifically in the "strain-specific DNA barcode" section it is unclear). What is the meaning of combined colors DNA? Why do you divide plant DNA by fungal DNA on the right side? You should think maybe about different scenarios as a supplementary figure to explain the different usages of the spikes. The current explanation in text + figure + legend is insufficient for understanding what is going on. Actually, the formulae provided in the materials and methods may be good to state explicitly in the figure along with the cartoon version. If you need space for that, the picture of the plant is not critical.
2. Figure 1C: to my eye, these shades of blue are quite similar, with a few outliers making the dynamic range of the values a bit hard to parse. I think a more wide ranging color palette would be better. In addition, what is the rationale behind the ordering of the letters (non-alphabetically)? It's not critical, but it is also not explained. The letters are not mentioned much overall, perhaps in figure 1d you can write "oligos a-j" next to the pool of oligos. The use of the word "position" here is confusing to me, because it's meant to describe the number of integrated fragments (correct?). The term "position" is not referenced in the legend, and I think should be either explained or changed to better reflect the meaning, because intuitively, I see it as somehow involved in position in the array of oligonucleotides. I think maybe size or something could be a better term.
3. The concept of spike DNA used for normalization may not be familiar to all readers. The fact that the same plasmid is used for chromosomal integration and as a spike is confusing, and I missed it when reading the paper for the first time. I suggest to: 1. Consider if this feature of the method should be included in the abstract (which currently ignores the spike role), 2.

7Explain the spike concept better in a figure and text (e.g. after line 170-171).

4. Figure 2. Figure 2A: what is spike 2? What is spike 1? I don't see it mentioned in the text. Maybe better to name them in a way that corresponds to what is meant to be normalized. I understand that the graph is just meant to show a linear relationship but still, it makes it hard to follow without telling us what the spike is. Further, I also think it is not necessary to show all the figures, maybe one representative and then dedicate more space to the rest of figure 2e-g, since it is a bit redundant to show similar linear relationships again and again. I like the comprehensive nature of the experiments that were done, but for presentation purposes, pick the best one and put the rest in supplementary. Figure 2E-F: I see the fact that there are different slopes and intercepts for each insert. I think something is missing for me in the description. It says kind of generally that there is heterogeneity in the lengths of the tags (lines 203-205). Is that the ultimate explanation for the heterogeneity? I think even though it's the results section, it might be worth spelling it out a bit more explicitly. Figure 2G: I think it would be good to show non-normalized vs. normalized bar graphs here, not just the normalized. The picture doesn't show me that the correction worked, other than the fact that the color bars should add up to be the size of the black bar (if I understood correctly), which is not visually easy to do without the colored bars being stacked one on top of the other. So I would perhaps do a stacked bar graph of the colors "before the correction", and then a stacked colored bar graph of "after the correction". Currently, the point is not so clearly shown visually.

5. Figure 4a-c. The legend mentions ellipses, but they are not in the figure. It makes it hard to interpret the p-values and their significance visually.

6. Lines 300-302: it is unclear where to observe a decrease in community composition difference. Please explain.

7. The application of the method is of course interesting. However, fig 5b, which shows the power of the spike, is highly similar to fig 5c which resulted from the data without the innovation of the method (simply normalizing by plant ITS). Maybe I misunderstood the analysis and the authors normalize to plant ITS from the spike. If this is the case I would like to see normalization to the plant ITS without the spike to see if the spike adds information.

8. Lines 349-351 I can't observe "a slight shift towards communities containing wild-type WCS358 alone". If this shift exists please indicate how you observe it.

9. Line 351-357: A genetic complementation of pqqf would be more informative than just mixing the two strains (wt and pqqf)

Minor revisions:

1. Line 264: should be fig 3B and not 2B.

2. Line 315: Extended figure 4, not extended figure 3

3. Line 320: I think this is meant to be a bold subtitle? There is no period at the end of the sentence.

4. Figure 6c-d: what is BC? It is not written in the legend

Wishing you best of luck!

Reviewer #3 (Remarks to the Author):

This is a substantial methods paper which outlines the development of a new technology for tagging individual strains of bacteria, then goes on to demonstrate its use to observe interactions between mutant strains and synthetic bacterial communities in Arabidopsis roots - observations that would have otherwise been impossible to make. These tools will be hugely useful for microbiome researchers in all fields, not only in plant microbiology.

I have some requests for clarification, changes or more discussion which should be addressed before publication, and I'll share those here.

On line 111: I suspect there have been other papers that have developed tag technology for bacteria, like for example in the study of the mouse microbiome. For example, look at: "High-throughput identification and quantification of single bacterial cells in the microbiota". Try see if there are other groups outside of plant microbiology that are doing similar work to yours? Describe how your approach is different.

On line 153 and 453: "core" is a loaded word. If you're going to use it, please define what you mean by it, like for instance, do these bacteria occur in all Arabidopsis roots under all natural conditions ever tested?

On line 215: Can you briefly speculate why there would variation in tag amplification from identical integration sites and in identical strains? Could this not be attributable to experimental error instead?

On line 287: Would it be possible to identify bacterial strains as such, rather than as "Root"? Maybe "Root Bacteria 418" or strain 418 instead? Was strain 418 deleted from the entire manuscript (please be explicit)? Why might this strain have died out and did you expect it to be such a poor survivor? Why was it included in the Syncom? Was strain 418 perhaps not really a "core" microbe? You could perhaps leave these matters for the discussion.

On line 320: Should this be a title in bold?

On line 339: You chose spike in over plant ITS, but you didn't really explain why - could you please state the advantages of one over the other?

On line 436: Please discuss why your spike in method results in fractional read counts (ie. less than one read). Is PCR off a spiked in plasmid much more efficient than a chromosomally integrated tag? Does this result in underestimating tag read counts?

In figure 2: Its a bit confusing to indicate corrected tags with this underlined title, while not saying anything about the 16S. Maybe organize this into a legend inside a box with a section for corrected and another section for uncorrected? Also the dotted lines don't appear to add any information, could you remove them?

In figure 3b: there's a faint band in the WT - I'm guessing that's contamination? Perhaps you should repeat the PCR to get a cleaner result.

In figure 5: Seeing multiple items in the legend labelled "Root" makes it seem like these were plant samples. Would it be possible to call these something like, "Root Isolate" or "Root

9Bacteria" or even just "Bacteria" or "Strain" to disambiguate?

Author Rebuttal to Initial comments

Point-to-point response letter to the comments and questions of the reviewers

Manuscript: "Simultaneous tracking of near-isogenic bacterial strains in synthetic *Arabidopsis* microbiota by chromosomally-integrated barcodes"

We would like to express our sincere appreciation for the reviewers' comments and constructive suggestions, which have helped us to improve our manuscript. We have performed additional experiments and modified the corresponding text sections to address all the referees' concerns.

To further show broad utility of the MoBacTag tool, the new Extended Data Fig. 7 demonstrates that MoBacTag-labeled strains can also be tracked in highly complex native communities of two soil types and a Jiffy peat matrix.

Reviewer #1:

This paper capitalizes on the widely-used Tn7 transposon in order to create a system to track closely-related bacterial strains in a community, and reveals some of the insights possible with such a system using gluconic acid mutants in *Pseudomonas*. The Tn7 transposon has an advantage over other transposons in that it has a predictable insertion site in those bacterial genomes in which it can integrate. The idea of using artificial genetic barcodes to distinguish closely related individuals is not novel, and the authors cite appropriate literature. What is novel is the particular design of the construct that contains the barcode, the design of the barcode itself, and the attention to robust validation of the results. Overall the paper is nicely written.

>>>>Major concerns

There are two major questions I have after reading this. The first is, can it work in my bacteria? The authors looked in their culture collection bioinformatically and found theoretical Tn7 insertion sites (glmS) in 98.6% of their genomes. Then in the paper, they use the Tn7 tags in *Pseudomonas* only. Other published papers using Tn7 for broad host range transformation attempt more strains <https://www.frontiersin.org/articles/10.3389/fmicb.2018.03052/full> and struggle with some of them.

10In this regard, it's a bit of an empty claim that all these genomes have insertion sites if it's not clear insertion will work in them. I would have really appreciated an estimate of the fraction of plant-associated bacterial genomes that can be transformed and transposed using standard techniques.

We thank the referee for her/his constructive comments. To show broad utility of the MoBacTag tool via *Tn7*-mediated integration, we have now tagged a total of 22 plant-derived bacterial strains from nine different genera (*Rhizobium*, *Neorhizobium*, *Pseudomonas*, *Xanthomonas*, *Lysobacter*, *Pseudoxanthomonas*, *Rhodanobacter*, *Achromobacter*, *Deinococcales*; line 159 in revised text). Bacterial methyl-specific restriction systems that recognize and degrade foreign DNA based on methylation patterns are critical barriers for successful transformation (e.g. Zhou *et al.*, 2012. *Current Microbiology*; <https://doi.org/10.1007/s00284-011-0048-5>). To further broaden the utility of our tagging system, we have generated a non-methylating, conjugation-competent *E. coli* (ET12467/pUZ8002) that replicates the restrictive *R6K* origin of replication used for MoBacTag vectors by chromosomal integration of *pir2* (line 176 in revised text). This additional biological material for the use of MoBacTag experiments is provided as a public resource.

As pointed out by the reviewer, some bacteria seem to be resistant to chromosomal integration mediated by *Tn7* (Schlechter *et al.*, 2018. *Frontiers in Microbiology*; <https://doi.org/10.3389/fmicb.2018.03052>). Due to the modular design of MoBacTag vectors, the *mini-Tn7* elements encoded in MoBacTag vectors can be easily exchanged with sequences for homologous recombination, thereby increasing the utility for different bacterial strains and possibly also fungi.

The other major question was related to the example case of the gluconic acid mutants. The authors presented intriguing results and mentioned that the *ppqF* mutant that could not make gluconic acid had a dominant phenotype in the bacterial community when mixed with wildtype *Pseudomonas*. They wrote “the immunosuppression mediated by gluconic acid synthesis in wild-type WCS358 is insufficient to support the establishment of wild type-like root communities in the presence of the WCS358:*ppqF* mutant.” But then, they do not explain why. Is it simply that the *ppqF* mutant competes for resources with the wild-type and then the wild-type isn’t abundant enough, and then for this reason the gluconic acid isn’t abundant enough? Also the *cyoB* mutant the authors mention had a mild effect compared to *ppqF*, and the authors suggest it could be (line 363) the “residual gluconic acid production in planta”, but then no followup. These were unsatisfying conclusions to the main biological message in this paper. Perhaps because this paper focuses on methods this could be said to be outside the scope of this work, but then this makes the first major concern above more relevant.

The referee is correct that we have not conducted follow-up experiments to investigate the mechanism underlying the dominant behavior of the WCS358:*ppqF* mutant. We proposed in the discussion that the WCS358:*ppqF* mutant might reduce the extracellular PQQ pool on roots due to lack of PQQ biosynthesis despite continuous PQQ consumption through WCS358:*ppqF* proliferation, resulting in PQQ deficiency in the bacterial community containing wild-type WCS358. As stated by the referee, the focus of our manuscript is indeed in providing a bacterial tagging tool suitable for the analysis of bacterial communities, including near-isogenic bacteria. To guide the reader already in the results section to a potential explanation for the dominant behavior of the WCS358:*ppqF* mutant, we have included the above hypothesis in the Results section (line 396 in revised text). In addition, we have added new data showing that chemical PQQ supplementation of the WCS358:*ppqF* mutant restored wild type-like D-glucose-dependent growth restriction *in vitro*, suggesting that WCS358 imports PQQ (line 398; Extended Data Figure 6).

One major question I had with the methods regards the differences in abundance for different MoBacTags in the same genetic background. The authors tested three tags in *Pseudomonas* (lines 2019 to 216). These data were collected using the 16S primers, which allowed also comparing tag abundance to 16S abundance. However, it is unclear from this whether the same tag-specific correction factors would also be needed if using the tag-specific primers instead of the 16S primers, for instance, or another primer set. Is there an intrinsic difference in the ability of each barcode to be amplified in PCR, or does it depend on what PCR primers are used?

Thank you. To directly address the question of whether differences in barcode DNA tag amplification

are specific for *16S rRNA* primers, we performed qPCR with dilution series of genomic DNA from MoBacTag-labeled wild-type WCS358, WCS358:pqqF and WCS358:cyoB using barcode-specific, plant *ITS*-specific and inter-gene chromosome-specific primers. The amplification efficiencies of the inter-gene chromosome-specific amplicon were comparable for wild-type WCS358, WCS358:pqqF and WCS358:cyoB, while we observed tag-specific amplification efficiencies with both the barcode-specific and plant *ITS*-specific primers (new Figure provided below for the reviewer and as extended data Figure x in the revised manuscript). Thus, differences in barcode DNA tag amplification are not limited to *16S rRNA* primers (line 238 in revised text).

>>>>Minor concerns

Line 206-210 “To exclude possible...

> It seems the authors are saying that they take three clones of WCS358 and put a different tag in the Tn7 site in each one. However, due to the wording about “the same 16S rRNA genetic background” etc, this is confusing. It can be worded more simply.

In the revised manuscript, we have simplified the sentence directly stating that we have tagged wild-type WCS358, WCS358:cyoB and WCS358:pqqF mutants (line 228 in revised text).

>Line 218 “homologous recombination”

Why use homologous recombination? Did Tn7 integration fail here?

Rhizobiales R13D was tagged either with a MoBacTag vector using *Tn7*-mediated integration (pBC61; used in Extended Data 3c) or MoBacTag vectors with sequences for homologous recombination. To show that barcode-specific amplification rates are independent of integration sites, we MoBacTag-labeled WCS358 strains with different barcode DNA tags at the same integration site downstream of *glmS* and tagged the Rhizobiales R13D by MoBacTag integration at different loci. We re-phrased the sentence accordingly (line 246 in revised manuscript).

Line 219 “Tag104”

> This tag is mentioned but not defined until the end of the sentence where it says “(termed tag104, tag93...” Best would be to say “We also generated and tested four variants of the root commensal Rhizobiales Root13D with MoBacTags (tag 104, tag93, and 94)” with the tags defined right after they are mentioned.

Thank you for the suggestion, which we have integrated in the revised text (line 248 in revised text).

Line 221 “DNA tag91”

> What is tag91 in? It wasn't referred to before.

We have mentioned tag91 now already in line 248 together with the other tags tested for *Rhizobium* R13D.

Line 223 “at identical and different chromosomal integration sites”

> How can something be identical and different? Please rewrite.

The reviewer is correct. We rephrased this sentence accordingly (line 254 in revised text).

Line 233-244 about tag correction factors..

> Interesting that each tag needs a correction factor. Can the authors speculate in the discussion why this may be the case?

Thank you for this comment.

Silverman *et al.* (2021. PLOS Computational Biology; <https://doi.org/10.1371/journal.pcbi.1009113>) reported PCR bias from ‘non-primer-mismatch sources’ in gut microbiota samples, representing multi-template PCRs. PCR on the gDNA of a MoBacTag-labeled strain are also multi-template PCRs with the DNA barcode tag and native *16S rRNA* sequence as templates. Aside from differences in sequence length, causes of bias from non-primer-mismatch sources in multi-template PCRs are still poorly understood (Silverman *et al.*, 2021), and we are not aware of systematic studies testing the influence of individual sequence properties on PCR biases from non-primer-mismatch sources. To prepare the reader for the observed amplification biases, we refer to the Silverman *et al.* 2021 study in the revised text (line 226).

Line 288

> Strain Root418 was less than 4% so the authors deleted it from analysis. Why? What’s the harm to leave it? How is this justified? How do the authors know it was irrelevant?

The reviewer is correct. We have now included R418 in all corresponding data analyses in the revised manuscript (see Figure below, line 323).

Figure 4d,e:Figure 5f,g:

Line 296 “which suggests..”

>It has been known for a long time that roots affect the microbes and have different communities than soil. So the use of the word “suggests” is not appropriate, because it looks like this is new information.

We agree with the referee and have replaced “suggests” with “confirming” in the revised manuscript (line 332 in revised manuscript).

Line 308 to 313:

> Says a few syncom members differed depending on treatment, but the results weren’t consistent. This paragraph makes one worried there is a problem, and starts giving details of individual strains that are affected, and then says the problem isn’t so bad and the results aren’t significant. Perhaps it can be simplified to just make the main point that differences observed were not significant.

The reviewer is correct that co-inoculation of WCS358:BC with the SynCoM does not result in a reproducible SynCom shift compared to co-inoculation with wild-type WCS358. Nevertheless, we wish to present these data in the interest of data transparency and to illustrate the importance of assessing the potential influence of an introduced antibiotic resistance encoded by a MoBacTag on a SynCom.

>FRT sites should work in theory, but not demonstrated. Would be nice to include guidance, at least, how to take advantage of them.

Thank you for the comment. We have added a guide to eliminate the antibiotic resistance from the MoBacTag using *FRT* sites in the Extended Data Methods (line 967 in revised text).

Line 331 to 337

>Can phrase in a way that shows the authors know how to interpret relative abundance. By saying “read counts... appeared to indicate enhanced colonization of *Streptomyces*.. “ it seems the authors

are admitting they got fooled, but it would be better to just say “read counts indicated a higher relative abundance of Streptomyces... which should not be misinterpreted as higher absolute abundance... Normalization of ...”

We have revised the manuscript as suggested by the referee (line 371 in revised text).

Line 336 “altered relative abundances... are likely an artefact”

> Relative abundances are just math.. they aren’t an artefact. Could say the relative abundances might incorrectly imply a change to absolute abundance or something like that.

This is a good point, we have now modified the sentence accordingly (line 376 in revised text).

Line 396 “essentially outcompeted”

> Can the authors use a number? What does this qualifier “essentially” actually mean?

We have now included that less than 10 reads for WCS358:cyoB in all 12 root and most (8/11) matrix samples, corresponding to a >100-fold reduction compared to wild-type WCS358 reads were detected (line 444 in revised text).

Line 434-436

> I know how a spike in works, but I am confused how it can “eliminate potential 454 barcode- and thereby sample-specific PCR bias”.

Thank you for the comment. If each sample is barcoded by a specific pair of 454-based barcodes with a putative intrinsic PCR bias, then amplicons of *16S rRNA*, barcode tags and the spike within one sample should have the same PCR bias mediated by the 454 barcode. If we then normalize using spike counts, we correct for this bias. We have modified the respective sentence in the revised manuscript to improve clarity (line 493)

Line 442 “single copy”

> In some genomes, there is more than one integration site, so this is not strictly true

Thank you for pointing this out; we have modified the text accordingly (line 507 in revised text).

Line 456-459

> This sentence was confusing.. please rewrite or split into two sentences.

We have split this sentence into two as suggested (line 521 in revised text).

Line 487 “tag-to-plant ITS”

>What drove the choice of a multiple copy number gene like plant ITS that differs between accessions vs. a single-copy plant gene for plant normalization?

Spike normalization used in our study allows for inter-sample comparisons but was not used here for absolute quantification. Several additional factors need to be considered to calculate absolute abundances. For instance, cell-type specific endoreduplication affects genome copy number per plant cell (Guo *et al.*, 2020. Plant Communications; <https://doi.org/10.1016/j.xplc.2019.100003>).

Our spike plasmid containing *16S rRNA* plus plant *ITS* primer binding sites enables studying the composition and load of microbial communities in experiments with multiple plant species. For example, we have recently observed host preference during the establishment of bacterial communities on roots of *A. thaliana* and *L. japonicus* (Wippel, Tao *et al.*, 2021. Nature Microbiology; <https://doi.org/10.1038/s41564-021-00941-9>).

>>>>FIGURES

1B

>Why is the synthetic DNA tag green and red, and everything else is just 1 color?

We wanted to indicate that the DNA barcode tag is specific for either strain A or B, i.e., it must be unique for one strain for each experiment. We now realize that this could be misleading and have changed the color in the revised manuscript to be specific for strain A.

2A

>Put the color legend in Figure 2A rather than 2D.. I couldn't find it at first

Thank you for your suggestion. We have moved the figure legend to the first panel showing linear correlations between spike concentration and read counts, which is Fig. 2b in the revised manuscript.

4 "Ellipses correspond to"

>There are no ellipses... a-c were not especially clear to me, and I wasn't sure exactly how to interpret, but perhaps the ellipses would help with that.

Thank you for pointing out this mishap. We have re-run the analysis to add the ellipses to Fig. 4a-c in

the revised manuscript.5 B and C

> This was a clear and consistent result showing importance of normalizing to the plant!

Thank you.

5D and E

> The colored boxes at the top showing condition 1 vs condition 2.. confusing meaning.. how do the colors relate to other things.. and “conditions” are not mentioned in the legend, so it’s hard to figure it out..it seems I compare condition 1 to condition 2 directly below and get the p-value shown? Please clarify in legend.

Response: We have changed “conditions” to “communities”, have added “(in presence of WCS358 derivative)” directly into the figure, and have added ellipses to the analysis. We have modified the figure legend accordingly.

“P values indicate statistical significance determined by permutational analysis of variance (PERMANOVA) test between the composition of synthetic community 1 and community 2 co-established with WCS358 derivatives indicated by colored boxes (permutation = 999). Ellipses correspond to Gaussian distributions fitted to each cluster (95% confidence interval).”

Reviewer #2:

Ordon, Thouin et al. propose a new technology, modular bacterial tags (MoBacTags), which can be used to quantitatively characterize the relative abundances of various bacteria at the resolution of the subspecies/strain. The quantitation ability of MoBacTags are calibrated and characterized, and are finally used in a proof-of-principle experiment with two *Pseudomonas* mutant strains of the same species (i.e. identical 16s sequences) which have known phenotypes. The design of the MoBacTag vector is smart as it combined multiple combinations of antibiotic marker genes, multiple combinations of fluorescent markers, and synthetic DNA tag that is flanked by multiple spikes which enable many types of data normalization. The authors invested thought and lab work in calibration to convince the reader that the method is fairly reliable. The checking of the method was thorough. Most of the paper is dedicated to the establishment of this method, and in this sense it is a bit too technical in our view, for a method that is interesting and useful but overall we think will not be a game changer in the plant microbiome field. A unique usage which the method enables is the mixture of multiple near-isogenic strains in the same experiment as was shown for example in the wt+pqf experiments. The effect of pqf deletion on the quantity of many members of the community is impressive to observe but for this result any spike that allows accurate quantification (and not just report a change in community composition as usually done in microbiome studies) can be used and the barcode was not critical if I understand it correctly.

We thank the referee for her/his constructive comments. We have incorporated the referee's suggestions and re-analyzed a subset of the data accordingly. We have also added missing details to improve clarity.

>>>>Major revisions

1. Figure 1A is critical for understanding the idea behind the method. Currently it is not clear enough. For example: it seems that "black DNA" refers to different things in every illustration (specifically in the "strain-specific DNA barcode" section it is unclear). What is the meaning of combined colors DNA? Why do you divide plant DNA by fungal DNA on the right side? You should think maybe about different scenarios as a supplementary figure to explain the different usages of the spikes. The current explanation in text + figure + legend is insufficient for understanding what is going on. Actually, the formulae provided in the materials and methods may be good to state explicitly in the figure along with the cartoon version. If you need space for that, the picture of the plant is not critical.

Thank you for the suggestions. As proposed by the reviewer, we have removed the plant symbol in the revised manuscript. We have also changed the potentially misleading term "bacterial/fungal spike plasmid" to "spike plasmid" and removed the "+" symbols. We replaced combined colors by ratios marked with "/". To increase consistency, we changed the color to depict the fungal ITS in 1b, 1d to distinguish it from the synthetic DNA barcode of the spike in Fig. 1a. Finally, we added labels to the sequences to explain the color code and illustrate the calculations for bacterial and plant abundance.

2. Figure 1C: to my eye, these shades of blue are quite similar, with a few outliers making the dynamic range of the values a bit hard to parse. I think a more wide ranging color palette would be better. In addition, what is the rationale behind the ordering of the letters (non-alphabetically)? It's not critical, but it is also not explained. The letters are not mentioned much overall, perhaps in figure 1d you can write "oligos a-j" next to the pool of oligos. The use of the word "position" here is confusing to me, because it's meant to describe the number of integrated fragments (correct?). The term "position" is not referenced in the legend, and I think should be either explained or changed to better reflect the meaning, because intuitively, I see it as somehow involved in position in the array of oligonucleotides. I think maybe size or something could be a better term.

We have changed the color palette of figure 1c and arranged the oligonucleotide identities in alphabetical order in the revised manuscript. We have now labeled the oligonucleotides in figure 1d with "oligo a-j". The term "position" refers indeed to the site of each oligonucleotide within the array rather than the length/size of the total DNA tag. We have examined how often a given oligonucleotide is found in a specific position in the forward or reverse orientation. To improve clarity, we have also indicated positions in Figure 1d above the oligonucleotide array. The figure legend was modified accordingly.

3. The concept of spike DNA used for normalization may not be familiar to all readers. The fact that the same plasmid is used for chromosomal integration and as a spike is confusing, and I missed it when reading the paper for the first time. I suggest to: 1. Consider if this feature of the method should be included in the abstract (which currently ignores the spike role), 2. Explain the spike concept better in a figure and text (e.g. after line 170-171).

Thank you for pointing this out. We have now visualized the spike plasmid in Figure 1b. Additionally, we have graphically illustrated the concept of spike normalization in Figure 2a and added an explanation of spike-based normalization in the revised manuscript (line 208). We have adjusted the figure legends accordingly. We believe it is not justified to mention this in the abstract as the method of spike-based normalization has been published and is not a main focus of our study.

Figure 1b:

Figure 2a:

4. Figure 2. Figure 2A: what is spike 2? What is spike 1? I don't see it mentioned in the text. Maybe better to name them in a way that corresponds to what is meant to be normalized. I understand that the graph is just meant to show a linear relationship but still, it makes it hard to follow without telling us what the spike is. Further, I also think it is not necessary to show all the figures, maybe one representative and then dedicate more space to the rest of figure 2e-g, since it is a bit redundant to show similar linear relationships again and again. I like the comprehensive nature of the experiments that were done, but for presentation purposes, pick the best one and put the rest in supplementary.

We have now moved the linear correlations for the BC and fungal ITS primers to the extended data figure, thereby creating space to introduce the concept of spike plasmids for normalization. We have also included the different uses of spike 1 and spike 2 in the text and have indicated their concentrations by a bar and a triangle below the x-axis (line 216 in revised text).

Figure 2E-F: I see the fact that there are different slopes and intercepts for each insert. I think something is missing for me in the description. It says kind of generally that there is heterogeneity in the lengths of the tags (lines 203-205). Is that the ultimate explanation for the heterogeneity? I think even though it's the results section, it might be worth spelling it out a bit more explicitly.

Thank you for this comment.

Silverman *et al.* (2021. PLOS Computational Biology; <https://doi.org/10.1371/journal.pcbi.1009113>) reported PCR bias from 'non-primer-mismatch sources' in gut microbiota samples, representing multi-template PCRs. PCR on the gDNA of a MoBacTag-labeled strain are also multi-template PCRs with the DNA barcode tag and native 16S rRNA sequence as templates. Aside from differences in sequence length, causes of bias from non-primer-mismatch sources in multi-template PCRs are still poorly understood (Silverman *et al.*, 2021), and we are not aware of systematic studies testing the influence of individual sequence properties on PCR biases from non-primer-mismatch sources. To prepare the reader for the observed amplification biases, we refer to the Silverman *et al.* 2021 study in the revised text (line 227).

Figure 2G: I think it would be good to show non-normalized vs. normalized bar graphs here, not just the normalized. The picture doesn't show me that the correction worked, other than the fact that the color bars should add up to be the size of the black bar (if I understood correctly), which is not visually easy to do without the colored bars being stacked one on top of the other. So I would perhaps do a stacked bar graph of the colors "before the correction", and then a stacked colored bar graph of "after the correction". Currently, the point is not so clearly shown visually.

Thank you for these suggestions, which facilitate comparison of corrected vs non-corrected read counts. We have replaced figure 2f to show stacked bar plots and included uncorrected tag abundances

in the revised manuscript. We modified the text accordingly to describe revised Figure 2f (line 257 in revised text).5. Figure 4a-c. The legend mentions ellipses, but they are not in the figure. It makes it hard to interpret the p-values and their significance visually.

Thank you for pointing out this mishap. We have re-analyzed the data and included ellipses.

6. Lines 300-302: it is unclear where to observe a decrease in community composition difference. Please explain.

To improve clarity, we now included that *p*-values are > 0.01 after *in silico* depletion of either only barcode-specific reads or additional depletion of *16S rRNA* sequences specific to the MoBacTag-labeled strain (line 338 in revised text).

7. The application of the method is of course interesting. However, fig 5b, which shows the power of the spike, is highly similar to fig 5c which resulted from the data without the innovation of the method

(simply normalizing by plant ITS). Maybe I misunderstood the analysis and the authors normalize to plant ITS from the spike. If this is the case I would like to see normalization to the plant ITS without the spike to see if the spike adds information.

Thank you for the comment. As suggested by the reviewer (see point 1), we have now included the calculation for plant ITS normalization in Fig. 1a (“ratio of bacteria to plant abundance”). We have further indicated the spike plasmid in Fig. 1b to highlight the two applications of the MoBacTag vectors. One application is as a spike plasmid for normalization and a second application is as a genomic DNA barcode to discriminate bacterial strains with identical *16S rRNA* sequences. Since *Arabidopsis* root samples colonized by a bacterial SynCom contain a vast excess of plant DNA, direct normalization of chromosomally integrated barcode reads by plant reads is not possible in our setup. Limitations in community profiling dominated by plant reads have also been described by Lundberg *et al.* (2021. *eLife*; <https://doi.org/10.7554/eLife.66186>). These authors bypassed this limitation by adjusting the ratio between plant and bacterial amplicons prior to sequencing and by additional deep sequencing of a reference sample for which the ratio was not adjusted. We have solved this complication through indirect normalization of bacterial to plant DNA as illustrated in Fig. 1a. The calculations for Fig. 5c are based on the same spike DNA amount relative to the total sample DNA amount for preparations of *16S* and plant ITS libraries. For the experiment shown in Fig. 5b,c, there is indeed no significant difference between the two analyses as we adjusted the total DNA concentrations for the preparation of the *16S* library.

Our spike plasmid containing *16S rRNA* plus plant ITS primer binding sites enables studying the composition and load of microbial communities in experiments with multiple plant species. For example, we have recently observed host preference during the establishment of bacterial communities on roots of *A. thaliana* and *L. japonicus* (Wippel, Tao *et al.*, 2021. *Nature Microbiology*; <https://doi.org/10.1038/s41564-021-00941-9>).

8. Lines 349-351 I can't observe “a slight shift towards communities containing wild-type WCS358 alone”. If this shift exists please indicate how you observe it.

We have re-analyzed the data and added ellipses in the revised manuscript in Fig 5d and 5e to highlight the community shifts. The *p*-value indicating differences between the communities established in the presence of wild-type WCS358 or WCS358:pqqF is lower (*p*-value=0.001) than the *p*-value indicating differences between communities established in the presence of wild-type WCS358 or wild-type WCS358 together with WCS358:pqqF (*p*-value=0.009). In panel 5e, the first axis is explained by co-inoculation of either wild-type WCS358 or WCS358:pqqF. Co-inoculations of combinations of wild-type WCS358 together with the mutants shift the communities to the left, i.e., closer to the community after co-inoculation of wild-type WCS358 only.

9. Line 351-357: A genetic complementation of *pqqf* would be more informative than just mixing the two strains (wt and *pqqf*)

We have not genetically complemented the WCS358::*pqqF* mutant. However, we chemically complemented this mutant by supplementing PQQ in the medium and observed wild type-like growth in a synthetic medium (Extended Data Fig. 6, line 398 in revised text).

>>>>Minor revisions:

1. Line 264: should be fig 3B and not 2B.

We have changed this accordingly.

2. Line 315: Extended figure 4, not extended figure 3

Thank you, we corrected this error.

3. Line 320: I think this is meant to be a bold subtitle? There is no period at the end of the sentence.

Indeed, this text was supposed to be a title.

4. Figure 6c-d: what is BC? It is not written in the legend

We have now replaced “BC read counts” with “barcode DNA tag read counts”.

Reviewer #3:

This is a substantial methods paper which outlines the development of a new technology for tagging individual strains of bacteria, then goes on to demonstrate its use to observe interactions between mutant strains and synthetic bacterial communities in *Arabidopsis* roots - observations that would have otherwise been impossible to make. These tools will be hugely useful for microbiome researchers in all fields, not only in plant microbiology.

I have some requests for clarification, changes or more discussion which should be addressed before publication, and I'll share those here.

We thank the referee for her/his comments. We have incorporated the referee's suggestions and comments in the revised manuscript.

On line 111: I suspect there have been other papers that have developed tag technology for bacteria, like for example in the study of the mouse microbiome. For example, look at: "High-throughput identification and quantification of single bacterial cells in the microbiota". Try see if there are other groups outside of plant microbiology that are doing similar work to yours? Describe how your approach is different.

Thank you for the suggestion. We have added several references from the vertebrate research field that examine tissue-specific bacterial pathogen population dynamics using barcode-tagged isogenic lines (line 111 in revised text). The therein described DNA barcodes were chromosomally-integrated into strain-specific loci using diverse methods like homologous recombination and CRISPR/Cas systems. However, the DNA barcodes itself were generally flanked with barcode-specific primers not allowing simultaneous community profiling. To improve clarity, we modified the text accordingly (line 119 in revised text). We suggest to not include the proposed publication by Jin *et al.*, 2022. We understand that this method quantifies bacterial cells and taxonomically classifies individual cells based on *16S rRNA* sequencing by barcoding isolated cells during library preparation. Since taxonomic classification is still based on differences in *16S rRNA* sequences, this method is insufficient to distinguish isogenic strains.

On line 153 and 453: "core" is a loaded word. If you're going to use it, please define what you mean by it, like for instance, do these bacteria occur in all *Arabidopsis* roots under all natural conditions ever tested?

Thank you for the comment. We have defined the word "core" in the revised manuscript at the position it is first mentioned and we cite references referring to the core microbiota of Angiosperm plant species (line 158 in revised text).

On line 215: Can you briefly speculate why there would variation in tag amplification from identical integration sites and in identical strains? Could this not be attributable to experimental error instead?

Thank you for the comment.

Silverman *et al.* (2021). PLOS Computational Biology; <https://doi.org/10.1371/journal.pcbi.1009113>) reported PCR bias from non-primer-mismatch sources in gut microbiota samples, representing multi-template PCRs. PCR on the gDNA of a MoBacTag-labeled strain are also multi-template PCRs with the DNA barcode tag and native *16S rRNA* sequence as templates. Aside from differences in sequence length, causes of bias from non-primer-mismatch sources in multi-template PCRs are still poorly understood (Silverman *et al.*, 2021), and we are not aware of systematic studies testing the influence of individual sequence properties on PCR biases from non-primer-mismatch sources. To prepare the reader for the observed amplification biases, we refer to the Silverman *et al.* 2021 study in the revised

text (line 227). According to Silverman *et al.* (2021), the relative abundance of two transcripts after PCR, here *16S rRNA* and barcode DNA tag, depends on the starting ratio of the two transcripts and the ratio of their amplification efficiencies. The starting ratio of *16S rRNA* to barcode DNA tag cannot be manipulated when sequencing pure cultures as it is an intrinsic factor of the genomic DNA. Thus, differences in relative transcript amounts after PCR must result from different amplification efficiencies.

On line 287: Would it be possible to identify bacterial strains as such, rather than as "Root"? Maybe "Root Bacteria 418" or strain 418 instead? Was strain 418 deleted from the entire manuscript (please be explicit)? Why might this strain have died out and did you expect it to be such a poor survivor? Why was it included in the Syncom? Was strain 418 perhaps not really a "core" microbe? You could perhaps leave these matters for the discussion.

Thank you for the comment. As suggested, we have replaced bacterial names with Rxxx, e.g. R418, and added an axis label "SynCom members" in Figure 4d and 4e and 5f and 5g to emphasize that these are bacteria.

In addition, we have now included R418 in all analysis. The SynCom design is explained in more detail in Wippel, Tao *et al.* (2021. *Nature Microbiology*; <https://doi.org/10.1038/s41564-021-00941-9>). Briefly, the strains were selected to be all distinguishable by V5-V7 *16S rRNA* sequences and to represent a diverse set of common bacterial families of the natural Arabidopsis root microbiota. The SynCom members were originally isolated from Arabidopsis roots of plants grown in a natural loamy soil. One possible explanation for the low abundance of R418 on roots in the Gnotopot plant growth system (Kremer *et al.*, 2021. *Nature Protocols*; <https://doi.org/10.1038/s41596-021-00504-6>) could be that R418 grows poorly in this peat-based matrix. Alternatively, R418 might generally be a low-abundance member of the root microbiota.

Figure 4d,e:

Figure 5f,g:

On line 320: Should this be a title in bold?

Thank you for pointing this out. The mishap was corrected accordingly in the revised manuscript.

On line 339: You chose spike in over plant ITS, but you didn't really explain why - could you please state the advantages of one over the other?

Thank you for pointing this out. As discussed in the revised manuscript, our spike plasmid containing 16S rRNA plus plant ITS primer binding sites enables studying the composition and load of microbial communities in experiments with multiple plant species (line 500). For example, we have recently observed host preference during the establishment of bacterial communities on roots of *A. thaliana* and *L. japonicus* (Wippel, Tao *et al.*, 2021. *Nature Microbiology*; <https://doi.org/10.1038/s41564-021-00941-9>).

On line 436: Please discuss why your spike in method results in fractional read counts (ie. less than one read). Is PCR off a spiked in plasmid much more efficient than a chromosomally integrated tag? Does this result in underestimating tag read counts?

Guo *et al.* (2020. *Plant Communications*; <https://doi.org/10.1016/j.xplc.2019.100003>) and Tkacz *et al.* (2018. *Microbiome*; <https://doi.org/10.1186/s40168-018-0491-7>) suggested that spike reads should account for 10%-60% and 20%-80% of the total reads per sample, respectively, as small numbers of spike reads could increase the variability of calibrated results. Normalization of reads from bacteria with low abundance, therefore, results in fractional read counts. Normalizations performed here allow for inter-sample comparisons but are not an absolute quantification. Several additional factors need to be considered to calculate absolute abundances. For instance, cell-type specific endoreduplication affects genome copy number per plant cell (Guo *et al.*, 2020. *Plant Communications*; <https://doi.org/10.1016/j.xplc.2019.100003>). In addition, DNA conformation is known to affect amplification efficiency, as systematically studied in Hou *et al.* (2010. *PLOS ONE* ; [10.1371/journal.pone.0009545](https://doi.org/10.1371/journal.pone.0009545)) or Lin *et al.* (2011. *PLOS ONE*; [10.1371/journal.pone.0029101](https://doi.org/10.1371/journal.pone.0029101)).

41In figure 2: Its a bit confusing to indicate corrected tags with this underlined title, while not saying anything about the 16S. Maybe organize this into a legend inside a box with a section for corrected and another section for uncorrected? Also the dotted lines don't appear to add any information, could you remove them?

Thank you for the suggestions. In response to a similar comment made by Reviewer #2, we have now included the uncorrected barcode reads in Fig. 2 to enable direct comparison of corrected and uncorrected read counts for each tag. We have also removed the dotted line as suggested.

In figure 3b: there's a faint band in the WT - I'm guessing that's contamination? Perhaps you should repeat the PCR to get a cleaner result.

Thank you for pointing this out. We have repeated the experiment and replaced Fig. 3b with newly obtained results.

In figure 5: Seeing multiple items in the legend labelled "Root" makes it seem like these were plant samples. Would it be possible to call these something like, "Root Isolate" or "Root Bacteria" or even just "Bacteria" or "Strain" to disambiguate?

Thank you for the suggestion. In line with Fig. 4d,e, we have shortened "Root" to "R" also in Fig. 5f and 5g and added an axis label "SynCom members" as suggested.

Decision Letter, first revision:

Message: Our ref: NMICROBIOL-23040945B

5th December 2023

Dear Dr. Schulze-Lefert,

Thank you for your patience as we've prepared the guidelines for final submission of your Nature Microbiology manuscript, "Simultaneous tracking of near-isogenic bacterial strains in synthetic *Arabidopsis* microbiota by chromosomally-integrated barcodes" (NMICROBIOL-23040945B). Please carefully follow the step-by-step instructions provided in the attached file, and add a response in each row of the table to indicate the changes that you have made. Please also check and comment on any additional marked-up edits we have proposed within the text. Ensuring that each point is addressed will help to ensure that your revised manuscript can be swiftly handed over to our production team.

43We would like to start working on your revised paper, with all of the requested files and forms, as soon as possible (preferably within two weeks). Please get in contact with us if you anticipate delays.

In recognition of the time and expertise our reviewers provide to Nature Microbiology's editorial process, we would like to formally acknowledge their contribution to the external peer review of your manuscript entitled "Simultaneous tracking of near-isogenic bacterial strains in synthetic *Arabidopsis* microbiota by chromosomally-integrated barcodes". For those reviewers who give their assent, we will be publishing their names alongside the published article.

Nature Microbiology offers a Transparent Peer Review option for new original research manuscripts submitted after December 1st, 2019. As part of this initiative, we encourage our authors to support increased transparency into the peer review process by agreeing to have the reviewer comments, author rebuttal letters, and editorial decision letters published as a Supplementary item. When you submit your final files please clearly state in your cover letter whether or not you would like to participate in this initiative. Please note that failure to state your preference will result in delays in accepting your manuscript for publication.

Cover suggestions

COVER ARTWORK: We welcome submissions of artwork for consideration for our cover. For more information, please see our [guide for cover artwork](https://www.nature.com/documents/Nature_covers_author_guide.pdf).

Nature Microbiology has now transitioned to a unified Rights Collection system which will allow our Author Services team to quickly and easily collect the rights and permissions required to publish your work. Approximately 10 days after your paper is formally accepted, you will receive an email in providing you with a link to complete the grant of rights. If your paper is eligible for Open Access, our Author Services team will also be in touch regarding any additional information that may be required to arrange payment for your article.

Please note that *Nature Microbiology* is a Transformative Journal (TJ). Authors may publish their research with us through the traditional subscription access route or make their paper immediately open access through payment of an article-processing charge (APC). Authors will not be required to make a final decision about access to their article until it has been accepted. <https://www.springernature.com/gp/open->

research/transformational-journals"> Find out more about Transformational Journals

Authors may need to take specific actions to achieve compliance with funder and institutional open access mandates. If your research is supported by a funder that requires immediate open access (e.g. according to Plan S principles) then you should select the gold OA route, and we will direct you to the compliant route where possible. For authors selecting the subscription publication route, the journal's standard licensing terms will need to be accepted, including self-archiving policies. Those licensing terms will supersede any other terms that the author or any third party may assert apply to any version of the manuscript.

For information regarding our different publishing models please see our Transformational Journals page. If you have any questions about costs, Open Access requirements, or our legal forms, please contact ASJournals@springernature.com.

Best regards,

Reviewer #1:

Remarks to the Author:

I am reviewer 1, and I was also asked to comment on whether the authors addressed comments of reviewer 2, who could not participate in this round.

In general I believe all comments from me and from reviewer 2 have been met. In particular, a wide variety of bacteria have been tagged, demonstrating more universality, and other points have been clarified. I have only 2 points to make for a short re-revision. I do not think it is necessary that I see the manuscript again. Whatever the authors and editor decide to do with the following comments will be adequate.

>>>Figure 1:

In figure 1a, panel 1, the picture shows a green and red bacterium overlapping each other and a single 16S DNA next to them. This makes sense, as both have the same 16S sequence so only one DNA molecule is shown.

In figure 1a, panel 3, the picture shows a green bacterium with a green tag divided by the spike, a red bacterium with a red tag divided by the spike, and an overlapping green and red bacterium with a single 16S DNA next to them. This also makes sense, as one gets three different DNA products.

However in figure 1b, panel 2, the picture shows a green bacterium with a green tag and a black 16S DNA. It also shows a red bacterium with a red tag and a black 16S DNA. Both 16S DNAs are black, but still there are 2 DNA molecules shown, such that 4 DNA molecules are present. I think this is confusing. Better to present it as in the third panel. That is, you will have a green bacteria with a green tag, a red bacterium with a red tag, and overlapped green and red bacteria with a single 16S DNA molecule. This keeps consistency with the other panels.

>>>Plant ITS normalization:

The role of the spike in and plant ITS is more clear to me with the updated Figure 1. Essentially you measure bacterial 16S relative to the spike in, and also measure plant ITS relative to the same spike in, and from this you can calculate the ratio of bacteria to plant.

I had asked the authors why they chose ITS instead of a single copy gene. I think one advantage of ITS is now more clear, in that one can use the same ITS primers on many different plant species, whereas single-copy genes are more likely to be unique to each plant species, and thus less universal as priming sites.

The authors said they used the spike in for inter-sample comparisons only. This would be valid for several Col-0 plants or Col-0 mutants, perhaps, but could cause problems if one wanted to compare the 5th leaf of Col-0 to the 5th leaf of Cvi-0, for example, because plant ITS copy number can be variable. I think it would be prudent to include some kind of warning to that effect to avoid people misusing the plant ITS normalization procedure in their experimental design.

Reviewer #3:

Remarks to the Author:

Thank you for addressing my comments. I am satisfied that ambiguities were explained and any issues corrected - I now endorse publication of this manuscript.

Author Rebuttal, first revision:

Point-to-point response letter: NMICROBIOL-23040945B

“Simultaneous tracking of near-isogenic bacterial strains in synthetic *Arabidopsis* microbiota by chromosomally-integrated barcodes” by Jana Ordon, Julien Thouin *et al.*,

We would like to thank the reviewers for their comments and suggestions on our revised manuscript. We have changed the corresponding text sections and figure to accommodate all comments.

Reviewer #1 (Remarks to the Author):

46I am reviewer 1, and I was also asked to comment on whether the authors addressed comments of reviewer 2, who could not participate in this round.

In general I believe all comments from me and from reviewer 2 have been met. In particular, a wide variety of bacteria have been tagged, demonstrating more universality, and other points have been clarified. I have only 2 points to make for a short re-revision. I do not think it is necessary that I see the manuscript again. Whatever the authors and editor decide to do with the following comments will be adequate.

>>>Figure 1:

In figure 1a, panel 1, the picture shows a green and red bacterium overlapping each other and a single 16S DNA next to them. This makes sense, as both have the same 16S sequence so only one DNA molecule is shown.

In figure 1a, panel 3, the picture shows a green bacterium with a green tag divided by the spike, a red bacterium with a red tag divered by the spike, and an overlapping green and red bacterium with a single 16S DNA next to them. This also makes sense, as one gets three different DNA products.

However in figure 1b, panel 2, the picture shows a green bacterium with a green tag and a black 16S DNA. It also shows a red bacterium with a red tag and a black 16S DNA. Both 16S DNAs are black, but still there are 2 DNA molecules shown, such that 4 DNA molecules are present. I think this is confusing. Beter to present it as in the third panel. That is, you will have a green bacteria with a green tag, a red bacterium with a red tag, and overlapped green and red bacteria with a single 16S DNA molecule. This keeps consistency with the other panels.

We have modified Figure 1a panel “strain-specific DNA barcode” as suggested by the reviewer.

>>>Plant ITS normalization:

The role of the spike in and plant ITS is more clear to me with the updated Figure 1. Essentially you measure bacterial 16S relative to the spike in, and also measure plant ITS relative to the same spike in, and from this you can calculate the ratio of bacteria to plant.

I had asked the authors why they chose ITS instead of a single copy gene. I think one advantage of ITS is now more clear, in that one can use the same ITS primers on many different plant species, whereas single-copy genes are more likely to be unique to each plant species, and thus less universal as priming sites.

The authors said they used the spike in for inter-sample comparisons only. This would be valid for several Col-0 plants or Col-0 mutants, perhaps, but could cause problems if one wanted to compare

the 5th leaf of Col-0 to the 5th leaf of Cvi-0, for example, because plant ITS copy number can be variable. I think it would be prudent to include some kind of warning to that effect to avoid people misusing the plant ITS normalization procedure in their experimental design.

Thank you for this suggestion. We have added a comment to the discussion (line 456): “However, differences in plant *ITS* copy number between plant species and accessions should be accounted for using correction factors.”

Reviewer #3 (Remarks to the Author):

Thank you for addressing my comments. I am satisfied that ambiguities were explained and any issues corrected - I now endorse publication of this manuscript.

Thank you for the positive feedback.

Final Decision Letter:

Message: 22nd January 2024

Dear Professor Schulze-Lefert,

Once again--I sincerely apologize for the delay. Despite the time it has taken, I am of course happy to be able to share good news: I am pleased to accept your Resource "Chromosomal barcodes for simultaneous tracking of near-isogenic bacterial strains in plant microbiota" for publication in Nature Microbiology. Thank you for having chosen to submit your work to us and many congratulations.

You may wish to make your media relations office aware of your accepted publication, in case they consider it appropriate to organize some internal or external publicity. Once your paper has been scheduled you will receive an email confirming the publication details. This is normally 3-4 working days in advance of publication. If you need additional notice of the

1date and time of publication, please let the production team know when you receive the proof of your article to ensure there is sufficient time to coordinate. Further information on our embargo policies can be found here:
<https://www.nature.com/authors/policies/embargo.html>

Please note that *Nature Microbiology* is a Transformative Journal (TJ). Authors may publish their research with us through the traditional subscription access route or make their paper immediately open access through payment of an article-processing charge (APC). Authors will not be required to make a final decision about access to their article until it has been accepted. [Find out more about Transformative Journals](https://www.springernature.com/gp/open-research/transformative-journals)

Authors may need to take specific actions to achieve [compliance](https://www.springernature.com/gp/open-research/funding/policy-compliance-faqs) with funder and institutional open access mandates. If your research is supported by a funder that requires immediate open access (e.g. according to [Plan S principles](https://www.springernature.com/gp/open-research/plan-s-compliance)) then you should select the gold OA route, and we will direct you to the compliant route where possible. For authors selecting the subscription publication route, the journal's standard licensing terms will need to be accepted, including [self-archiving policies](https://www.nature.com/nature-portfolio/editorial-policies/self-archiving-and-license-to-publish). Those licensing terms will supersede any other terms that the author or any third party may assert apply to any version of the manuscript.

An online order form for reprints of your paper is available at <https://www.nature.com/reprints/author-reprints.html>. All co-authors, authors' institutions and authors' funding agencies can order reprints using the form

appropriate to their geographical region.

With kind regards,